# Addition of a carboxy-terminal tail to the normally tailless gonadotropin-releasing hormone receptor impairs fertility in female mice

Chirine Toufaily[1], Jérôme Fortin[1], Carlos AI Alonso[1], Evelyne Lapointe[2], Xiang Zhou[1], Yorgui Santiago-Andres[3], Yeu-Farn Lin[1], Yiming Cui[1], Ying Wang[1], Dominic Devost[1], Ferdinand Roelfsema[4], Frederik Steyn[5,6], Aylin C Hanyaloglu[7], Terence E Hébert[1], Tatiana Fiordelisio[3], Derek Boerboom[2], Daniel J Bernard[1]*

[1]Department of Pharmacology and Therapeutics, McGill University, Montreal, Canada; [2]Département de biomédecine vétérinaire, Université de Montréal, Ste-Hyacinthe, Canada; [3]Departamento de Ecología y Recursos Naturales, Biología, Facultad de Ciencias, Universidad Nacional Autónoma de México, Ciudad Universitaria, Mexico City, Mexico; [4]Department of Internal Medicine, Section Endocrinology and Metabolic Diseases, Leiden University Medical Center, Leiden, Netherlands; [5]School of Biomedical Sciences, Faculty of Medicine, The University of Queensland, Brisbane, Australia; [6]Department of Neurology, Royal Brisbane & Women's Hospital, Queensland, London, United Kingdom; [7]Institute of Reproductive and Developmental Biology, Department of Metabolism, Digestion and Reproduction, Imperial College, London, United Kingdom

*For correspondence:
daniel.bernard@mcgill.ca

Competing interest: The authors declare that no competing interests exist.

**Abstract** Gonadotropin-releasing hormone (GnRH) is the primary neuropeptide controlling reproduction in vertebrates. GnRH stimulates follicle-stimulating hormone (FSH) and luteinizing hormone (LH) synthesis via a G-protein-coupled receptor, GnRHR, in the pituitary gland. In mammals, GnRHR lacks a C-terminal cytosolic tail (Ctail) and does not exhibit homologous desensitization. This might be an evolutionary adaptation that enables LH surge generation and ovulation. To test this idea, we fused the chicken GnRHR Ctail to the endogenous murine GnRHR in a transgenic model. The LH surge was blunted, but not blocked in these mice. In contrast, they showed reductions in FSH production, ovarian follicle development, and fertility. Addition of the Ctail altered the nature of agonist-induced calcium signaling required for normal FSH production. The loss of the GnRHR Ctail during mammalian evolution is unlikely to have conferred a selective advantage by enabling the LH surge. The adaptive significance of this specialization remains to be determined.

## Editor's evaluation

The authors have studied the effects of addition of a C tail to mammalian GnRH receptor. This is very well conducted study with very nicely designed experiments and appropriate conclusions. This is an interesting study that was well conducted and written. It is an important observation on how evolution of the GnRH receptors have contributed to reproductive processes and hence an important study in the area of reproductive biology.

## Introduction

The propagation and survival of all species depend on reproduction. In vertebrates, the process is controlled by hormones in the hypothalamic–pituitary–gonadal axis. Arguably, the hypothalamic decapeptide gonadotropin-releasing hormone (GnRH) is the most important brain hormone regulating reproduction (*Whitlock et al., 2019*; *Maggi et al., 2016*; *Brown and Roberson, 2017*; *Conn et al., 1987*). Disruption of GnRH synthesis, secretion, or action can delay or prevent puberty or cause infertility. GnRH acts via its receptor, GnRHR, in pituitary gonadotrope cells. GnRHR agonists and antagonists are used clinically in assisted reproductive technologies and to treat hormone-dependent diseases (*Maggi et al., 2016*; *Huirne and Lambalk, 2001*; *Huerta-Reyes et al., 2019*; *Corona et al., 2017*).

GnRH is released in pulses from neuron terminals in the median eminence into the pituitary portal vasculature. The hormone binds GnRHR on the plasma membrane of gonadotropes, stimulating the synthesis and secretion of the gonadotropins, luteinizing hormone (LH) and follicle-stimulating hormone (FSH) (*Bouligand et al., 2009*; *Wolczynski et al., 2003*; *Costa, 2001*; *Topaloglu et al., 2006*; *Cattanach et al., 1977*). LH and FSH are heterodimeric glycoproteins composed of the gonadotropin α subunit (product of the *Cga* gene) noncovalently linked to hormone-specific β-subunits: LHβ (*Lhb*) or FSHβ (*Fshb*), respectively (*Combarnous, 1988*; *Pierce and Parsons, 1981*; *Cahoreau et al., 2015*). GnRH stimulates the expression of all three gonadotropin subunit genes (*Thompson et al., 2013*; *Thompson and Kaiser, 2014*; *Kaiser et al., 1997*) as well as its own receptor (*Gnrhr*) (*Kaiser et al., 1997*; *Loumaye and Catt, 1982*; *Hazum and Keinan, 1982*; *Clayton et al., 1980*).

LH and FSH regulate gonadal function, most notably steroidogenesis and gamete maturation (*Layman, 2000*; *Abel et al., 2000*; *Kumar et al., 1997*). Gonadal sex steroids negatively feedback to the hypothalamus to control their own synthesis by inhibiting GnRH secretion (*Smith et al., 2005b*; *Smith et al., 2005a*; *Pielecka-Fortuna et al., 2008*). In addition, in females, in the late follicular phase of the menstrual cycle in primates or in the afternoon of proestrus in the rodent estrous cycle, high estrogen levels stimulate GnRH secretion through positive feedback, generating a high amplitude, long duration surge of LH, which triggers ovulation (*Richards et al., 1998*).

The type 1 GnRHR is a rhodopsin-like G-protein-coupled receptor (GPCR) (*Sealfon et al., 1997*). Remarkably, in mammals, GnRHR lacks the intracellular carboxyl-tail (Ctail) that is characteristic of most GPCRs, including GnRHRs in nonmammalian vertebrates like birds, amphibians, and fish (*Blomenröhr et al., 2002*). The Ctail plays important roles in GPCR function. Agonist binding to many GPCRs leads to receptor internalization and homologous desensitization (*Sun and Kim, 2021*). These processes are often mediated by the phosphorylation of the Ctail by G-protein receptor kinases, recruitment of adaptor proteins such as β-arrestins 1 and 2, and receptor endocytosis via a clathrin-dependent pathway (*Hilger et al., 2018*; *Magalhaes et al., 2012*). Upon ligand binding, the mammalian GnRHR is not phosphorylated, does not recruit arrestins, and is internalized slowly and with poor efficiency (*Vrecl et al., 2000*; *Castro-Fernández and Conn, 2002*; *Willars et al., 1999*; *Davidson et al., 1994*). Thus, the mammalian GnRHR is not subject to homologous desensitization in the conventional sense. As a result, the receptor has the potential to continue signaling during times of protracted GnRH secretion, as occurs prior to ovulation. Some have speculated, therefore, that the loss of the Ctail during evolution may have conferred an ability to the mammalian GnRHR to broker long duration, high-amplitude LH surges (*Davidson et al., 1994*; *Perrett and McArdle, 2013*), but this was never before addressed directly in vivo. It is notable, however, that LH surges are observed in nonmammalian vertebrates with GnRHRs containing Ctails, such as birds (*Liu et al., 2001*). Moreover, GnRHRs lacking Ctails have been observed in some nonmammalian vertebrates (*Williams et al., 2014*; *Sefideh et al., 2014*).

To gain greater insight into the potential significance of the loss of the Ctail in the mammalian GnRHR, we generated a knockin mouse model that expresses a chimeric GnRHR in which the chicken GnRHR Ctail was fused in frame to the C-terminus of the murine GnRHR. Importantly, the addition of a chicken Ctail altered, but did not prevent LH surges. Unexpectedly, the data provide novel insight into mechanisms of GnRH-stimulated FSH synthesis.

## Results

### Generation of knockin mice expressing a chimeric murine/chicken GnRHR

Using gene targeting in embryonic stem cells, we generated knockin mice in which the endogenous exon 3 of *Gnrhr* was replaced by a modified exon 3 encoding the C-terminus of murine GnRHR fused in-frame with the intracellular Ctail of the chicken GnRHR (*Figure 1—figure supplement 1A, B*). Heterozygous mice (*Gnrhr*$^{Ctail/+}$) were interbred to produce wild-type (WT, *Gnrhr*$^{+/+}$), heterozygous (*Gnrhr*$^{Ctail/+}$), and homozygous (Ctail, *Gnrhr*$^{Ctail/Ctail}$) animals, which were born at the expected Mendelian frequencies (*Figure 1—figure supplement 1C*). Ctail mice expressed an mRNA encoding the chimeric receptor in their pituitaries (*Figure 1—figure supplement 1D*).

### Ctail mice are hypogonadal and subfertile

We assessed the reproductive function of female and male Ctail mice relative to their WT littermates. When paired with WT C57BL/6 males, Ctail females produced smaller litters than WT (*Figure 1A*). A minority of Ctail mice were profoundly subfertile or infertile. Ctail females exhibited fewer estrous cycles per week (*Figure 1B*), due to an extended amount of time spent in estrus (*Figure 1—figure supplement 2*). Ovarian mass was reduced in Ctail females relative to WT (*Figure 1C, D*), but uterine weight was not significantly altered (*Figure 1E*). The numbers of antral follicles (*Figure 1F*), preovulatory follicles (*Figure 1G*), and corpora lutea (*Figure 1H*) were reduced in Ctail relative to WT ovaries, indicating impairments in folliculogenesis and ovulation. Ctail males were hypogonadal (*Figure 1I, J*) and oligozoospermic (*Figure 1K*), but their seminal vesicle masses were comparable to those of WT (*Figure 1I, L*).

### Serum FSH levels are reduced in Ctail mice

To help explain the observed hypogonadism in Ctail mice, we next examined gonadotropin secretion. In females sampled on diestrus afternoon, serum FSH and LH levels did not differ significantly (two-way analysis of variance) between genotypes ('sham' data in *Figure 2A, B*), though there was a clear trend for reduced FSH in Ctail mice. Indeed, the difference was significant when analyzed directly after the removal of the confirmed outlier in the WT group [$t(19) = 2.1$, $p = 0.0012$, two-tailed]. A second cohort of females was ovariectomized (OVX) to remove gonadal hormone (steroids and inhibin) feedback and increase endogenous GnRH secretion. Under these conditions, FSH and LH levels increased, as expected (*Figure 2A, B*). There was no significant genotype difference observed, but both gonadotropins trended lower in OVX Ctail relative to WT females.

In males, serum FSH levels were significantly reduced in gonad-intact ('sham') Ctail relative to WT mice (*Figure 2C*). In contrast, both single time point (*Figure 2D*) and pulsatile LH release were statistically normal in Ctail males (*Figure 2—figure supplement 1*). FSH levels did not increase postcastration, but the difference between genotypes persisted (*Figure 2C*). The postcastration increase in LH levels was blunted in Ctail relative to WT males (*Figure 2D*).

### Pituitary gonadotropin subunit and *Gnrhr* mRNA levels are altered in Ctail mice

To better understand the reduced gonadotropin levels in Ctail mice, we evaluated pituitary gonadotropin subunit (*Fshb*, *Lhb*, and *Cga*) and *Gnrhr* expression. In gonad-intact ('sham') animals, *Fshb* mRNA levels were significantly reduced in male, but not in female Ctail relative to WT mice (*Figure 3A, E*). Following gonadectomy, *Fshb* mRNA levels were increased in both genotypes, but the response was blunted in Ctail mice, significantly so in males (with a clear trend in females) (*Figure 3A, E*). Similar to serum LH levels, pituitary expression of the *Lhb* and *Cga* subunits did not differ between gonad-intact WT and Ctail males and females (*Figure 3B, C, F, G*). In contrast, following gonadectomy, increases in *Lhb* and *Cga* expression were significantly blunted in Ctail mice (*Figure 3B, C, F, G*), paralleling the observed effects on LH secretion (*Figure 2B, D*). *Gnrhr* mRNA expression was significantly reduced in gonad-intact female and male Ctail relative to WT mice (*Figure 3D, H*). Following gonadectomy, the difference between genotypes was no longer statistically significant, but levels continued to trend lower in Ctail mice.

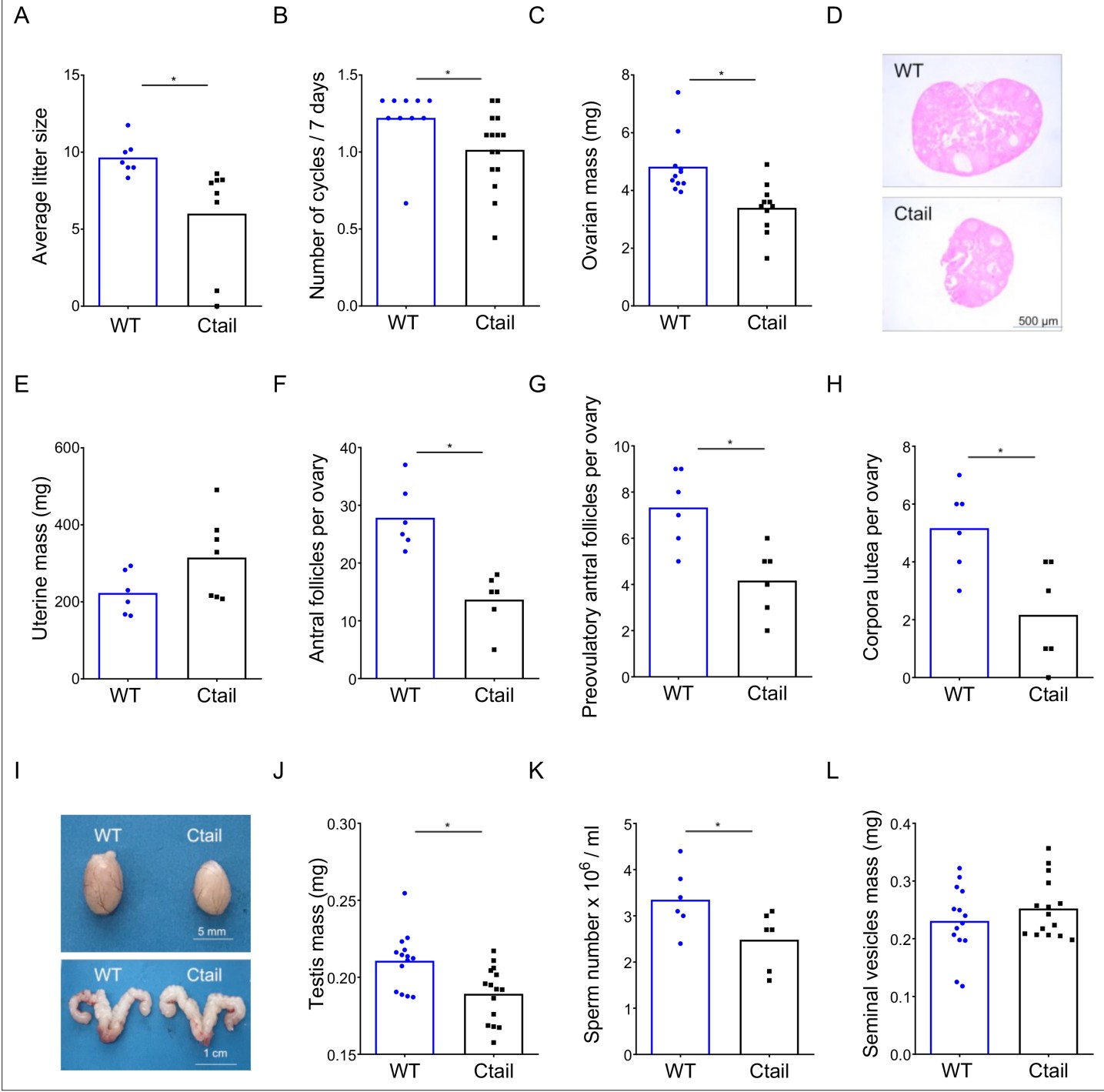

**Figure 1.** Ctail mice are hypogonadal and subfertile. (**A**) Average litter sizes in wild-type (WT) and Ctail females paired with WT C57BL/6 males over a 6-month breeding trial (*p = 0.0196). (**B**) Estrous cycle frequency in WT and Ctail females (*p = 0.0384). (**C**) Ovarian mass of 10- to 12-week-old females (*p = 0.002). (**D**) H&E-stained ovarian sections from a WT and a Ctail mouse. (**E**) Uterine mass of 10- to 12-week-old females (ns, p = 0.088). Numbers of (**F**) antral follicles (*p = 0.0008), (**G**) preovulatory follicles (*p = 0.0055), and (**H**) corpora lutea per ovary (*p = 0.0088). (**I**) Testes (top) and seminal vesicles (bottom) from a WT and a Ctail male. (**J**) Testicular mass (*p = 0.0037), (**K**) number of mature spermatozoa per testis (*p = 0.0468), and (**L**) seminal vesicle mass (ns, p = 0.3123) in 10- to 12-week-old males. In A–C, E–H, and J–L, the bar height reflects the group mean and dots and squares reflect individual animals. Student's *t*-tests were performed for statistical analysis.

The online version of this article includes the following figure supplement(s) for figure 1:

**Figure supplement 1.** Generation of Ctail mice by gene targeting in embryonic stem cells.

**Figure supplement 2.** Ctail females exhibit altered estrous cyclicity.

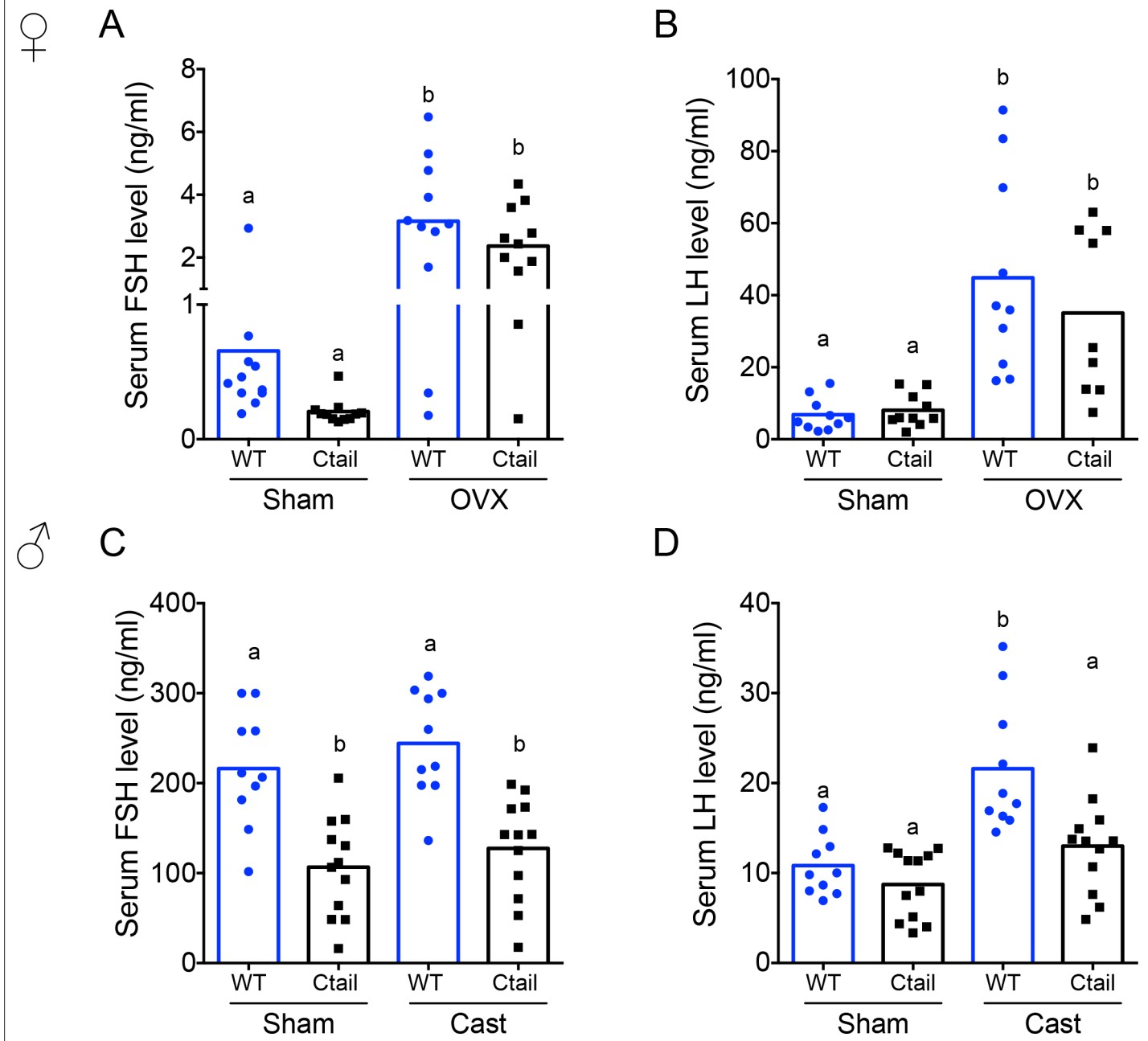

**Figure 2.** Serum follicle-stimulating hormone (FSH) levels are reduced in Ctail mice. Serum (**A, C**) FSH and (**B, D**) luteinizing hormone (LH) levels were measured in 10- to 12-week-old sham-operated (Sham) or gonadectomized female (**A, B**) and male (**C, D**) wild-type (WT) and Ctail mice. Females were sampled on diestrus afternoon. Gonadectomized animals were sampled 2-week postovariectomy (OVX) or castration (Cast). Male serum samples were measured with FSH/LH multiplex assays. In females, FSH was measured using an FSH Luminex assay and LH levels in females were measured by in-house ELISA. In each panel, the bar height reflects the group mean and dots and squares reflect individual animals. Statistical analyses in all panels were performed using two-way analyses of variance (ANOVAs), followed by Tukey's multiple comparison tests. Bars with different letters differed significantly [female FSH: WT (sham) vs. Ctail (sham) p = 0.8241; WT (sham) vs. WT (OVX) p = 0.0001; Ctail (sham) vs. Ctail (OVX) p = 0.0010; WT (OVX) vs. Ctail (OVX) p = 0.4372. Female LH: WT (sham) vs. Ctail (sham) p = 0.99861; WT (sham) vs. WT (OVX) p = 0.0002; Ctail (sham) vs. Ctail (OVX) p = 0.0126; WT (OVX) vs. Ctail (OVX) p = 0.6399; male FSH: WT (sham) vs. Ctail (sham) p = 0.0005; WT (sham) vs. WT (Cast) p = 0.7155; Ctail (sham) vs. Ctail (Cast) p = 0.8218; WT (Cast) vs. Ctail (Cast) p = 0.0002. Male LH: WT (sham) vs. Ctail (sham) p = 0.9162; WT (sham) vs. WT (Cast) p = 0.0002; Ctail (sham) vs. Ctail (Cast) p = 0.2477; WT (Cast) vs. Ctail (Cast) p = 0.0018].

The online version of this article includes the following figure supplement(s) for figure 2:

**Figure supplement 1.** Normal luteinizing hormone (LH) pulse frequency in male Ctail mice.

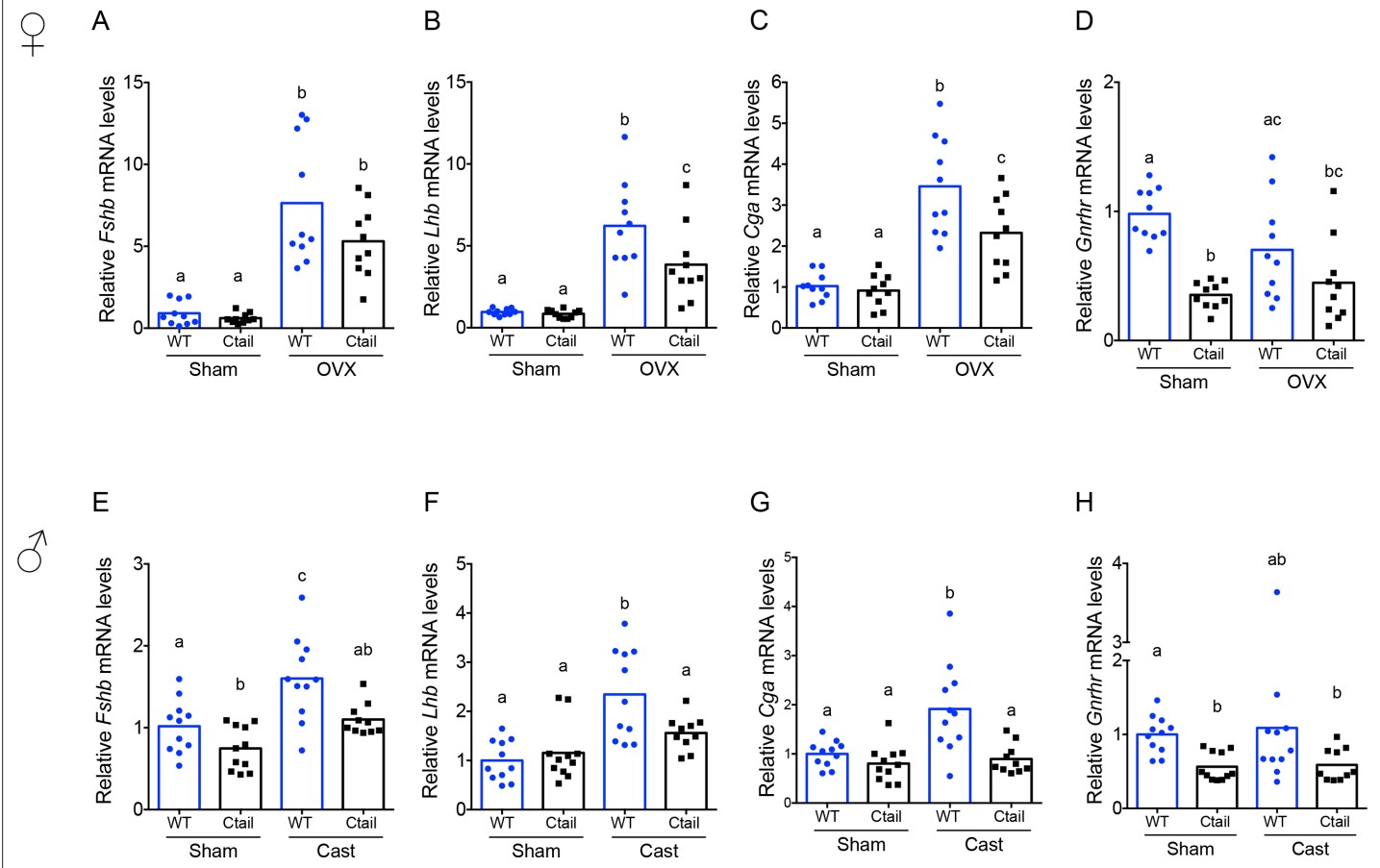

**Figure 3.** Pituitary gonadotropin subunit and *Gnrhr* mRNA levels are regulated by genotype and gonadal status. Relative pituitary (**A, E**) *Fshb*, (**B, F**) *Lhb*, (**C, G**) *Cga*, and (**D, H**) *Gnrhr* mRNA levels in the mice from **Figure 2** were measured by RT-qPCR. Gene expression was normalized to the reference gene ribosomal protein L19 (*Rpl19*). In each panel, the bar height reflects the group mean and dots and squares reflect individual animals. Statistical analyses were performed using two-way analysis of variance (ANOVA) tests, followed by Tukey's multiple comparison test. Bars with different letters differed significantly [female *Fshb*: wild-type (WT; sham) vs. Ctail (sham) p = 0.99; WT (sham) vs. WT (OVX) p < 0.0001; Ctail (sham) vs. Ctail (OVX) p = 0.0002; WT (OVX) vs. Ctail (OVX) p = 0.1057]. Female *Lhb*: WT (sham) vs. Ctail (sham) p = 0.9989; WT (sham) vs. WT (OVX) p < 0.0001; Ctail (sham) vs. Ctail (OVX) p = 0.0033; WT (OVX) vs. Ctail (OVX) p = 0.0267. Female *Cga*: WT (sham) vs. Ctail (sham) p = 0.9893; WT (sham) vs. WT (OVX) p < 0.0001; Ctail (sham) vs. Ctail (OVX) p = 0.0016; WT (OVX) vs. Ctail (OVX) p = 0.0138. Female *Gnrhr*: WT (sham) vs. Ctail (sham) p < 0.0001; WT (sham) vs. WT (OVX) p = 0.1332; Ctail (sham) vs. Ctail (OVX) p = 0.8731; WT (OVX) vs. Ctail (OVX) p = 0.1922. Male *Fshb*: WT (sham) vs. Ctail (sham) p = 0.0468; WT (sham) vs. WT (Cast) p = 0.0019; Ctail (sham) vs. Ctail (Cast) p = 0.1103; WT (Cast) vs. Ctail (Cast) p = 0.0109. Male *Lhb*: WT (sham) vs. Ctail (sham) p = 0.9345; WT (sham) vs. WT (Cast) p < 0.0001; Ctail (sham) vs. Ctail (Cast) p = 0.4394; WT (Cast) vs. Ctail (Cast) p = 0.0266. Male *Cga*: WT (sham) vs. Ctail (sham) p = 0.8192; WT (sham) vs. WT (Cast) p = 0.0013; Ctail (sham) vs. Ctail (Cast) p = 0.9772; WT (Cast) vs. Ctail (Cast) p = 0.0005. Male *Gnrhr*: WT (sham) vs. Ctail (sham) p = 0.0029; WT (sham) vs. WT (Cast) p = 0.6634; Ctail (sham) vs. Ctail (Cast) p > 0.9999; WT (Cast) vs. Ctail (Cast) p = 0.5638.

## The preovulatory LH surge is blunted in female Ctail mice

GnRH secretion is increased postgonadectomy (because of the loss of steroid negative feedback) and at the time of the LH surge (because of steroid positive feedback). Given impairments in LH production and/or release in gonadectomized Ctail mice, we next examined the naturally occurring preovulatory surge on the afternoon of proestrus. Five of eight WT females (62.5%) exhibited clear LH surges, while no Ctail mice (0% of *N* = 7) surged on proestrus, as determined by vaginal smearing (**Figure 4A, B**). As estrous cyclicity was altered in Ctail mice, we reasoned that we might have missed surges that actually occurred. Therefore, we took a different approach to assess natural LH surges, which previously proved successful in our assessment of LH surges in gonadotrope-specific progesterone receptor knockouts (**Toufaily et al., 2020**). Blood samples were collected four times daily over 11 consecutive days. Over the sampling interval, we detected LH surges in 93% of WT mice compared to 43% of Ctail animals (**Figure 4C** and **Figure 4—figure supplement 1**). The maximal LH levels

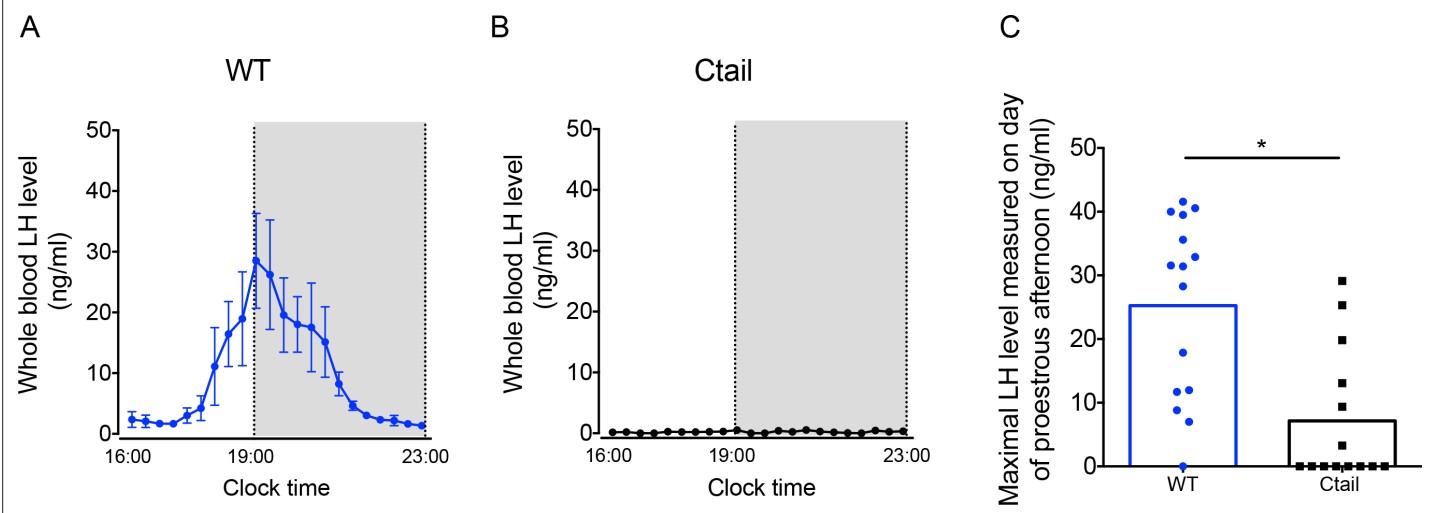

**Figure 4.** Luteinizing hormone (LH) surge amplitude is attenuated in Ctail females. (**A**) Preovulatory LH surge profiles in wild-type (WT; $N = 5$) and (**B**) Ctail females ($N = 5$) on proestrus, as identified by vaginal cytology. Blood samples were collected every 20 min from 4:00 pm (16:00) to 11:00 pm (23:00). Gray areas represent the dark phase of the light/dark cycle. (**C**) Maximal LH levels measured on proestrus from WT and Ctail females sampled four times daily for 11 days (see *Methods*). LH levels were measured in whole blood by with an in-house ELISA. In panels A and B, each dot reflects the group mean ± standard error of the mean (SEM). In C, the bar height reflects the group mean and dots and squares reflect individual animals. A Student *t*-test was performed for statistical analysis, *p = 0.0006.

The online version of this article includes the following figure supplement(s) for figure 4:

**Figure supplement 1.** Ctail females exhibit altered luteinizing hormone (LH) surges.

measured were significantly blunted in Ctail relative to WT females (*Figure 4C*). The timing of the surge did not appear to differ between genotypes (*Figure 4—figure supplement 1*).

## LH release is impaired in Ctail mice following GnRH stimulation

Blunted LH release both postgonadectomy and during the proestrus surge suggested that GnRH action in gonadotropes might be altered in Ctail mice. To more directly assess GnRH responsiveness, we performed GnRH stimulation tests in vivo. Mice of both genotypes released LH in response to exogenous GnRH, with peaks observed 15-min postinjection and returning to baseline by 1 hr (*Figure 5A, D*). However, the amplitude of the response was blunted in Ctail relative to WT mice. Intrapituitary FSH and LH levels were lower in female Ctail relative WT littermates (*Figure 5B, C*). In contrast, in males, pituitary FSH content did not differ between genotypes (*Figure 5E*), but pituitary LH content was slightly higher in Ctail than WT males (*Figure 5F*).

## GnRH activation of G$\alpha_{q/11}$ via the Ctail receptor is impaired in vitro

The reductions in FSH production under basal conditions and in LH release when GnRH secretion was enhanced suggested that GnRH signaling was somehow altered in gonadotropes of Ctail mice. The GnRHR is canonically coupled to G$\alpha_{q/11}$ (*Naor, 2009*). We therefore interrogated G$\alpha_{q/11}$-dependent signaling downstream of WT and Ctail forms of the murine GnRHR in vitro. As assessed using a G$\alpha_q$ bioluminescence resonance energy transfer (BRET)-based biosensor, GnRH-dependent G$\alpha_q$ activation was markedly attenuated in heterologous HEK 293 cells expressing the Ctail relative to WT GnRHR receptor (*Figure 6A*), even though cell surface expression of the two receptors was equivalent (*Figure 6—figure supplement 1*).

G$\alpha_{q/11}$ signaling is associated with activation of phospholipase C, which cleaves phosphatidylinositol 4,5-bisphosphate into diacylglycerol (DAG) and inositol 1,4,5-trisphosphate (IP$_3$). As revealed with a DAG BRET-based biosensor, GnRH-dependent DAG production was impaired in HEK 293 cells expressing the Ctail compared to WT receptor (*Figure 6B*). GnRH stimulation of IP$_1$ production (a surrogate for IP$_3$) was also significantly attenuated in HEK 293 cells expressing the Ctail GnRHR (*Figure 6C*).

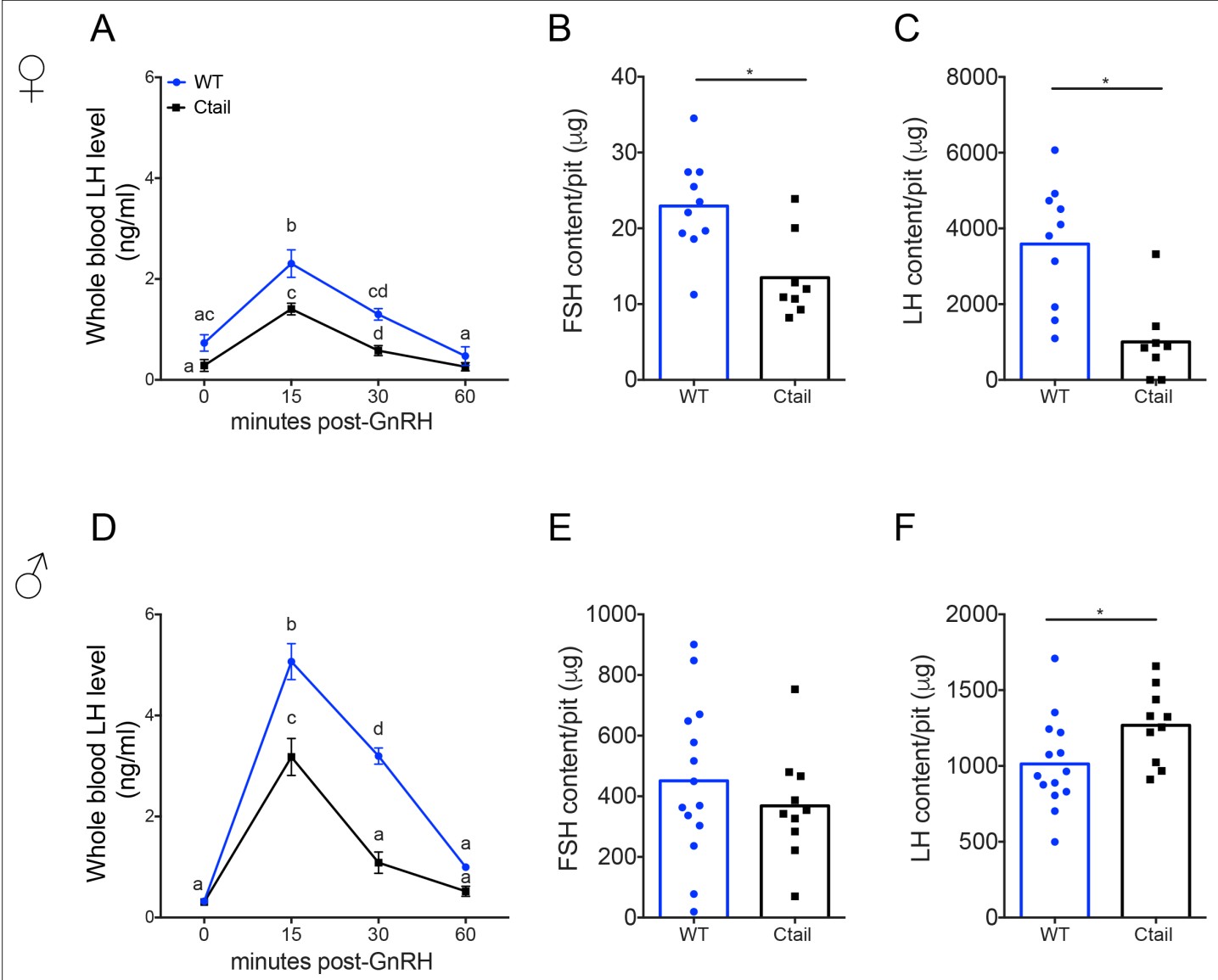

**Figure 5.** Gonadotropin-releasing hormone (GnRH)-stimulated luteinizing hormone (LH) release is attenuated in Ctail mice. Whole blood LH levels in 10- to 12-week-old (**A**) female and (**D**) male wild-type (WT; blue, $N = 12$ females and $N = 14$ males) and Ctail (black, $N = 9$ for females and $N = 11$ for males) mice before (0) and 15-, 30-, and 60-min post-i.p. injection of 1.25 ng of GnRH per g of body mass. Each point is the mean ± standard error of the mean (SEM). Data were analyzed using two-way analyses of variance (ANOVAs), followed by Tukey's post hoc tests for multiple comparisons. Points with different letters differ significantly (females WT vs. Ctail: 0 min $p = 0.0514$; 15 min $p = 0.0139$; 30 min $p = 0.0002$, 60 min $p = 0.3536$; males WT vs. Ctail: 0 min $p = 0.959653$; 15 min $p < 0.0001$; 30 min $p < 0.0001$, 60 min $p = 0.1112$). Intrapituitary contents of (**B, E**) follicle-stimulating hormone (FSH) and (**C, F**) LH in randomly cycling (**B, C**) female and (**E, F**) male WT and Ctail mice. The bar height reflects the group mean and dots and squares reflect individual animals. Data were analyzed by Student's $t$-tests (B, *$p = 0.0043$; C, *$p = 0.0122$; E, *$p = 0.3977$; F, *$p = 0.0394$). FSH levels were measured using a Luminex assay in females and by RIA in males. LH levels were measured with the in-house ELISA in both sexes.

## GnRH-dependent calcium mobilization, but not extracellular regulated kinase signaling, is impaired downstream of the Ctail receptor in HEK 293 cells

DAG, alone or in combination with calcium, activates protein kinase C (PKC) isoforms (*Oliva et al., 2005*). PKC, in turn, activates mitogen-activated protein kinase signaling (*Naor et al., 2000*). GnRH activation of the extracellular regulated kinase 1/2 (ERK1/2) pathway is particularly important for *Lhb* transcription (*Bliss et al., 2009*; *Yamada et al., 2004*; *Harris et al., 2002*). In the gonadotrope-like cell line, LβT2, GnRH induction of ERK1/2 phosphorylation (pERK1/2) is Gα$_q$ (*Figure 7—figure*

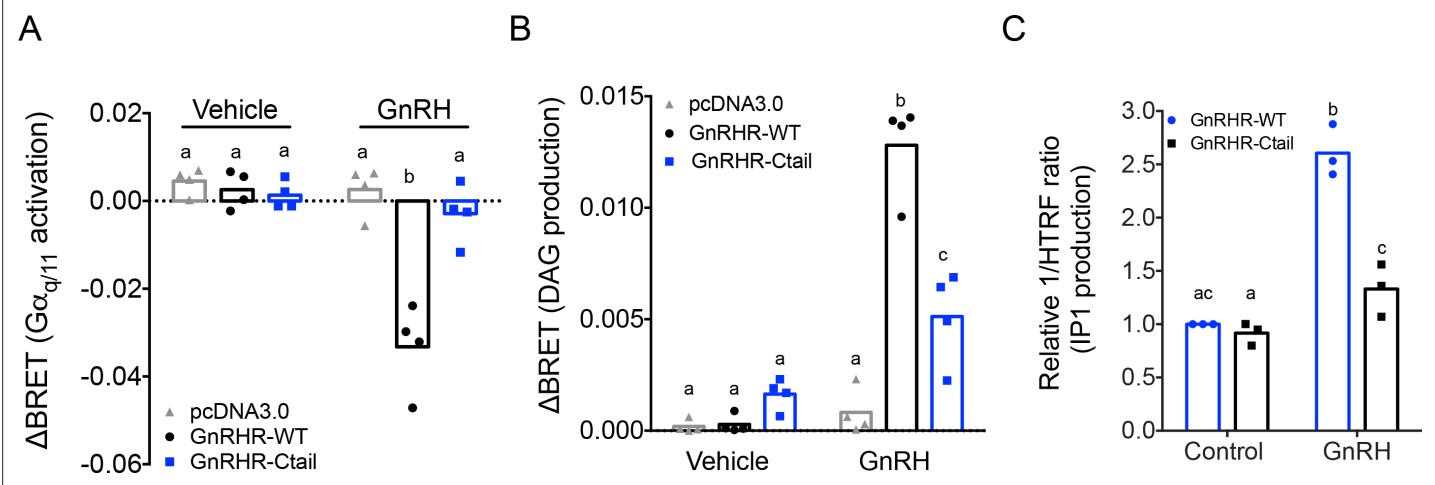

**Figure 6.** Gonadotropin-releasing hormone (GnRH) activation of $G_q$, diacylglycerol, and inositol phosphate via GnRHR-Ctail is impaired in heterologous cells. HEK 293 cells were transfected with empty vector (pcDNA3.0), GnRH-WT, or GnRHR-Ctail with (**A**) $G_q$ or (**B**) DAG BRET-based biosensors. Cells were loaded with Coelenterazine 400a for 5 min, and luminescence values recorded 10 s before and 30-s post-treatment with vehicle (water) or 100 nM GnRH. ΔBRET values were calculated as the average of BRET values before treatment minus the average values post-treatment. Data are shown from four independent experiments. Bar heights reflect group means. (**C**) HEK 293 cells were transfected with GnRHR-WT or GnRHR-Ctail. Cells were treated with vehicle or 100 nM GnRH for 30 min. IP1 production was measured and reported as represented as the inverse of the Homogeneous Time-Resolved Fluorescence (HTRF) ratio relative to control condition. Data are shown from three independent experiments. Bar heights reflect group means. In all panels, two-way analyses of variance (ANOVAs) followed by Tukey's multiple comparison test was used for statistical analysis. Different letters indicate statistically significant differences. In (**A**), untreated vs. treated with GnRH: pcDNA3.0 p = 0.9969; GnRHR-WT p < 0.0001; and GnRHR-Ctail p = 0.9093. In GnRH-treated conditions: pcDNA3.0 vs. GnRHR-WT p < 0.0001; pcDNA3.0 vs. GnRHR-Ctail p = 0.7775, and GnRHR-WT vs. GnRHR-Ctail p < 0.000, and in (**B**), untreated vs. treated with GnRH: pcDNA3.0 p = 0.9838; GnRHR-WT p < 0.0001 and GnRHR-Ctail p = 0.0180. In GnRH-treated conditions: pcDNA3.0 vs. GnRHR-WT p < 0.0001; pcDNA3.0 vs. GnRHR-Ctail p = 0.0029, and GnRHR-WT vs. GnRHR-Ctail p < 0.0001. In (**C**), GnRHR-WT (control) vs. GnRHR-Ctail (control) p = 0.6321; GnRHR-WT (control) vs. GnRHR-WT (GnRH) p = 0.0004; GnRHR-Ctail (control) vs. GnRHR-Ctail (GnRH) p = 0.0170; GnRHR-WT (GnRH) vs. GnRHR-Ctail (GnRH) p = 0.0073.

The online version of this article includes the following source data and figure supplement(s) for figure 6:

**Figure supplement 1.** Cell surface expression of wild-type (WT) and Ctail receptors is comparable in HEK 293T cells.

**Figure supplement 1—source data 1.** Source data for *Figure 6—figure supplement 1A*.

supplement 1A, B) and PKC dependent (*Figure 7—figure supplement 1C, D*; *Larivière et al., 2007*; *Naor et al., 1998*), but calcium independent (*Figure 7—figure supplement 1E, F*). In transfected HEK 293 cells, GnRH-induced pERK1/2 equivalently via the WT and Ctail receptors (*Figure 7A, B*) and in both cases was PKC dependent (*Figure 7C, D*). Although the Ctail receptor acquired the ability to recruit β-arrestin-1 (*Figure 7—figure supplement 2A, B*), GnRH activation of ERK1/2 signaling was arrestin independent (*Figure 7—figure supplement 2C, D*).

GnRH induction of intracellular calcium mobilization, which depends on $IP_3$, was reduced in Ctail relative to WT GnRHR expressing cells (*Figure 7E*). This impairment was not caused by the Ctail receptor's enhanced internalization, as the defect was not rescued in cells lacking arrestins (*Figure 7—figure supplement 2E*).

## GnRH-dependent calcium signaling is altered in gonadotropes of GnRHR-Ctail mice

As the above analyses were conducted in heterologous cells, we next examined GnRH regulated calcium signaling in gonadotropes of adult male WT and Ctail mice using a whole pituitary ex vivo preparation (see Methods). As expected, the three well-characterized GnRH-induced calcium response patterns were observed in individual gonadotropes of WT mice (*Figure 8A*): oscillatory (*Figure 8B*), biphasic (*Figure 8C, D*), and transient (*Figure 8E*). In contrast, gonadotropes of Ctail mice showed more uniform responses to GnRH (*Figure 8F*), with extended oscillatory or biphasic patterns that were not seen in WT (*Figure 8G–J*). There was no significant difference in the area under the curve (AUC) between genotypes (*Figure 8K*), or the maximum intensity of response (MIF, *Figure 8L*);

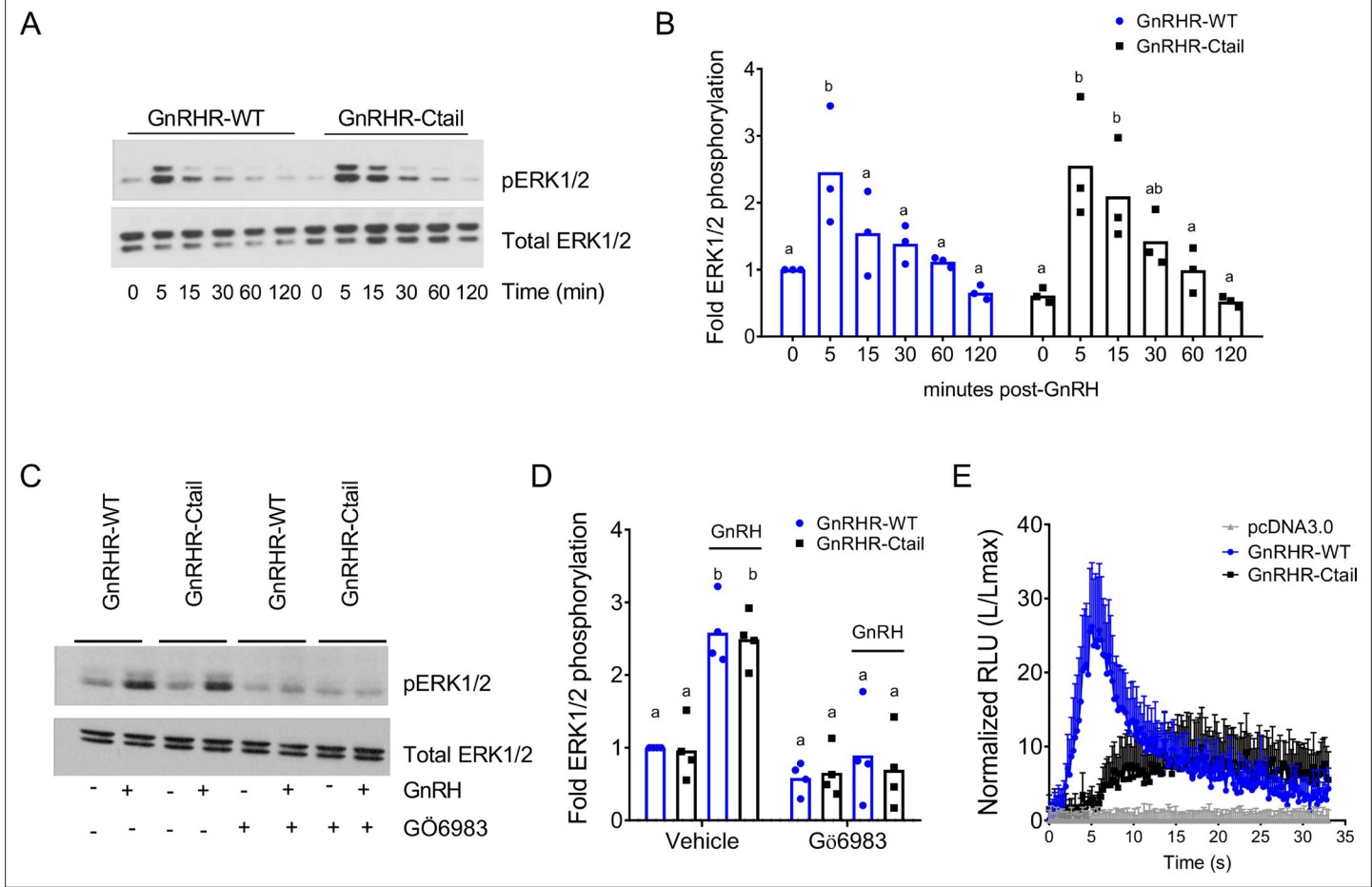

**Figure 7.** Gonadotropin-releasing hormone (GnRH)-stimulated intracellular calcium mobilization, but not ERK1/2 phosphorylation, is attenuated via GnRH-Ctail in heterologous cells. (**A**) HEK 293 cells were transfected with GnRH-WT or GnRHR-Ctail. Twenty-four hours post-transfection, cells were treated with vehicle (water; 0) as control or 100 nM GnRH for 5, 15, 30, 60, and 120 min. Whole cell protein lysates were collected and subjected to sodium dodecyl sulphate–polyacrylamide gel electrophoresis (SDS–PAGE) and western blotting with phospho- (top) or total (bottom) ERK1/2 antibodies. Blots from 1 of 3 replicate experiments are shown. (**B**) Data from the three independent experiments exemplified in panel A were quantified by normalizing the densitometry for pERK1/2 to total ERK1/2 and are presented relative to the control condition of the WT receptor. Two-way analysis of variance (ANOVA) followed by Tukey's multiple comparison test was used for statistical analysis. Bars with different letters differed significantly. GnRH-WT: 0 vs. 5 min p = 0.0179; 0 vs. 15 min p = 0.7673; 0 vs. 30 min p = 0.9309; 0 vs. 120 min p = 0.9572; GnRH-Ctail: 0 vs. 5 min p = 0.0010; 0 vs. 15 min p = 0.0156; 0 vs. 30 min p = 0.3846; 0 vs. 120 min p > 0.9999. (**C**) HEK 293 cells were transfected with GnRH-WT or GnRHR-Ctail. Twenty-four-hour post-transfection, cells were pretreated with 5 μM pan-PKC inhibitor GÖ6983 for 20 min, and then treated with vehicle (water) or 100 nM GnRH for 5 min. Western blotting was performed as in panel A. One blot from four independent experiments is presented. (**D**) Data from the four independent experiments exemplified in panel C were quantified and statistically analyzed as in panel B. In GnRH-treated conditions: GnRHR-WT (vehicle) vs. GnRHR-Ctail (vehicle) p > 0.9999; GnRHR-WT (GÖ6983) vs. GnRHR-Ctail (GÖ6983) p = 0.9969; GnRHR-WT (vehicle) vs. GnRHR-WT (GÖ6983) p = 0.0001; GnRHR-Ctail (vehicle) vs. GnRHR-Ctail (GÖ6983) p < 0.0001. (**E**) HEK 293 cells were transfected with GnRHR-WT, GnRHR-Ctail, or empty vector (pcDNA3.0) along with the luminescence Obelin biosensor. Twenty-four-hour post-transfection, cells were loaded with Coelenterazine cp for 2 hr. Cells were then treated with 100 nM GnRH. Intracellular Ca²⁺ was measured as relative luminescence emitted every 22 ms over 0.5 min. Data are presented as the ratio of total luminescence after GnRH over maximal luminescence (not shown) following Triton X-100 treatment from three independent experiments (mean ± standard error of the mean [SEM]).

The online version of this article includes the following source data and figure supplement(s) for figure 7:

**Source data 1.** Source data for *Figure 7A*.

**Source data 2.** Source data for *Figure 7C*.

**Figure supplement 1.** Gonadotropin-releasing hormone (GnRH)-induced ERK1/2 phosphorylation is $G\alpha_{q/11}$ and protein kinase C (PKC) dependent, and calcium independent in homologous LβT2 cells.

**Figure supplement 1—source data 1.** Source data for *Figure 7—figure supplement 1A*.

**Figure supplement 1—source data 2.** Source data for *Figure 7—figure supplement 1C*.

*Figure 7 continued on next page*

Figure 7 continued

**Figure supplement 1—source data 3.** Source data for *Figure 7—figure supplement 1E*.

**Figure supplement 2.** Gonadotropin-releasing hormone (GnRH)-induced pERK1/2 and calcium mobilization via GnRHR-Ctail are β-arrestin independent.

**Figure supplement 2—source data 1.** Source data for *Figure 7—figure supplement 2A*.

however, the number of peaks per cell was significantly higher in Ctail gonadotropes (*Figure 8M*). When we correlated the AUC with the MIF, we observed a difference in the mobilization of calcium between genotypes, with Ctail gonadotropes having lower amplitude but longer duration intracellular calcium elevations (*Figure 8N*). WT gonadotropes showed principally oscillatory or biphasic response patterns, whereas Ctail gonadotropes exhibited more extended responses (*Figure 8O*). The transient response pattern occurred in fewer than 1% of cells and therefore was not examined quantitatively.

We further examined calcium response patterns to repeated GnRH pulses. Gonadotropes of both genotypes responded to a second GnRH pulse, 1 hr after the first, with no evidence of desensitization in either case (*Figure 8—figure supplement 1*). Indeed, the responses to the second pulse were comparable to the first in terms of AUC (*Figure 8—figure supplement 1B, E*), MIF (*Figure 8—figure supplement 1C, F*), and oscillatory patterns (*Figure 8—figure supplement 1G–M*). Next, we asked to what extent the response patterns depended on influx of calcium via voltage-gated L-type channels. As expected, the L-type channel blocker nimodipine altered GnRH responses in gonadotropes of WT animals, reducing both the AUC and MIF, as previously reported in *Kwiecien and Hammond, 1998*; *Figure 8—figure supplement 2A–C*. Nimodipine also reduced AUC and MIF in gonadotropes of the Ctail mice (*Figure 8—figure supplement 2D–F*). GnRH-induced calcium oscillations observed in control gonadotropes were absent in the presence of nimodipine (*Figure 8—figure supplement 2G–I*). Interestingly, the prolonged GnRH-induced calcium oscillations (both oscillatory and biphasic responses) were decreased considerably in nimodipine-treated Ctail gonadotropes and, in most cells, were no longer present (*Figure 8—figure supplement 2J–N*).

## GnRH induction of *Fshb* expression is dependent on intracellular calcium

The mechanisms through which GnRH induces *Fshb* expression are poorly understood. However, given the impairments in FSH production in Ctail mice (*Figures 2 and 3*) and altered profile of GnRH-induced calcium signaling via the Ctail receptor (*Figures 7E and 8*), we asked whether there is a role for calcium in GnRH regulation of *Fshb*. A single pulse of GnRH was sufficient to induce *Fshb*, but not *Lhb* mRNA levels in LβT2 cells (*Figure 9A, B*). This is not unlike the situation in GnRH-deficient mice (*hpg*), where once daily GnRH is sufficient to induce FSH but not LH production (*McDowell et al., 1982a*; *McDowell et al., 1982b*). GnRH-induced *Fshb* expression in LβT2 cells was blocked by the calcium chelator, BAPTA-AM (*Figure 9A*), which did not affect basal *Lhb* mRNA levels (*Figure 9B*). Pulsatile GnRH is required for LH induction in GnRH-deficient mice (*Gibson et al., 1991*). Pulsatile GnRH stimulated both *Fshb* and *Lhb* mRNA expression in LβT2 cells and these responses were blocked with BAPTA-AM (*Figure 9C, D*), but not nimodipine (*Figure 9—figure supplement 1*).

## Not all Ctails impair G$\alpha_q$ activation and calcium mobilization via chimeric GnRHRs

Finally, we asked whether the effects observed with the chicken Ctail on the murine GnRHR occur with other non-mammalian GnRHR C-tails. Therefore, we added C-tails from type II GnRHRs of *Xenopus laevis* (frog), *Anolis carolinensis* (lizard), or *Clarias garepinus* (catfish) to the murine GnRHR (*Figure 10—figure supplement 1*). The *Anolis* C-tail impaired GnRH-induced G$\alpha_q$ activation and calcium mobilization in HEK 293 cells, as observed with the chicken C-tail (*Figure 10A, B*, black and purple). In contrast, GnRH signaled via the *Clarias* chimeric receptor in a manner indistinguishable from the WT murine GnRHR (*Figure 10A, B*, green and blue). Addition of the *Xenopus* C-tail modestly attenuated GnRH-induced G$\alpha_q$ activation but had no effect on calcium mobilization (*Figure 10A, B*, pink). GnRH effectively induced ERK1/2 phosphorylation via all of the chimeric receptors in a PKC-dependent manner (*Figure 10C, D*).

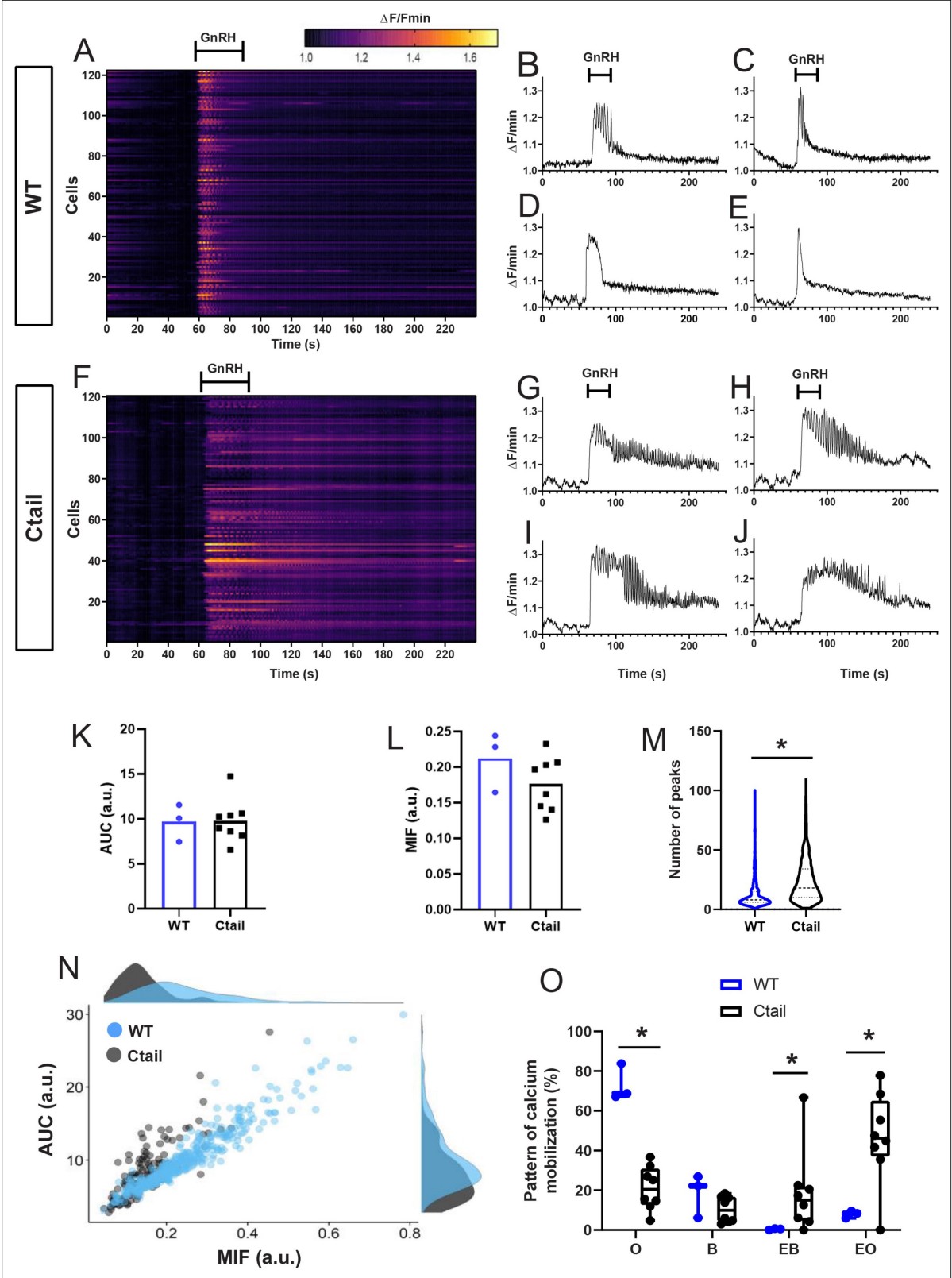

**Figure 8.** Gonadotropin-releasing hormone (GnRH)-stimulated calcium responses are altered in gonadotropes of Ctail mice. Raster plots of calcium responses in gonadotropes from a representative adult male (**A**) wild-type (WT) and (**F**) Ctail mouse. Each row represents an individual cell. Cells are numbered on the *y*-axis. The *x*-axis shows time in seconds. The timing of GnRH administration is indicated. The heatmap at the top shows the strength of the response, with yellower colors reflecting stronger calcium responses. Examples of calcium responses in individual gonadotropes from (**B–E**) WT

*Figure 8 continued on next page*

*Figure 8 continued*

and (**G–J**) Ctail mice. (**K**) Area under the curve (AUC) and (**L**) maximum intensity of fluorescence (MIF) were quantified (*N* = 3 WT and *N* = 8 Ctail). (**M**) Numbers of oscillatory peaks were also quantified (p < 0.0001; *N* = 3 [710 cells] WT and *N* = 8 [534 cells] Ctail). (**N**) Correlation between AUC vs. MIF measurements from one mouse per genotype (250 cells of WT and 128 cells of Ctail). (**O**) Comparison of the calcium pattern of response (*N* = 3 WT and *N* = 8 Ctail): oscillatory (O) (73.223 ± 7.489 vs. 21.037 ± 10.262; WT vs. Ctail, respectively, p = 0.0121); biphasic (B) (18.388 ± 8.885 vs. 10.430 ± 6.075; WT vs. Ctail, respectively, p = ns); extended biphasic (EB) (0.419 ± 0.307 vs. 21.444 ± 19.528; WT vs. Ctail, respectively, p = 0.0167); and extended oscillatory (EO) (7.969 ± 1.497 vs. 53.036 ± 14.069; WT vs. Ctail, respectively, p = 0.0167). Statistical analyses were performed using Wilcoxon rank-sum tests. *, significantly different.

The online version of this article includes the following figure supplement(s) for figure 8:

**Figure supplement 1.** Gonadotropes of wild-type (WT) and Ctail mice respond to repeated gonadotropin-releasing hormone (GnRH) pulses.

**Figure supplement 2.** Nimodipine alters gonadotropin-releasing hormone (GnRH)-induced calcium responses in gonadotropes of wild-type and Ctail mice.

## Discussion

The loss of the carboxy-terminal tail from the GnRH receptor during mammalian evolution was previously hypothesized to be an adaptation that enabled preovulatory LH surges (*Willars et al., 1999*; *Pawson et al., 2008*). The data presented here challenge this idea. Addition of the chicken GnRHR Ctail to the endogenous murine GnRHR blunted but did not block the LH surge. With few exceptions, females expressing the mouse-chicken chimeric GnRHR (GnRHR-Ctail) were fertile, but with smaller litter sizes compared to WT mice. Reductions in FSH rather than perturbations of the LH surge likely explain their subfertility. The FSH impairment appears to derive from alterations in GnRH-induced calcium signaling.

## Effects of the chicken Ctail on gonadotropin synthesis and secretion

Serum FSH and pituitary *Fshb* mRNA levels are lower in GnRHR-Ctail than WT mice. In males, this is associated with small, but significant decreases in testis mass and spermatogenesis. There is a direct relationship between Sertoli cell number and spermatogenic potential (*Griswold, 1998*). Sertoli cell number is regulated by FSH during early postnatal development in rodents (*Kumar et al., 1997*; *Allan et al., 2004*). Though we did not quantify Sertoli cells in GnRHR-Ctail males or their FSH levels prior to weaning, it seems likely that the FSH deficiency observed in adulthood also occurs earlier in life in these animals. Indeed, depleting FSH in young but not adult mice reduces testis size and sperm counts (*Kumar et al., 1997*; *Li et al., 2018*). In females, reduced FSH levels are associated with decreased numbers of preovulatory follicles. Because most GnRHR-Ctail females exhibit LH surges and/or corpora lutea, it is clear that the majority could and did ovulate. Therefore, the most parsimonious explanation for the subfertility in these females is impaired follicle development secondary to FSH deficiency.

Though present, LH surges are altered in most GnRHR-Ctail females. Unfortunately, we were unable to fully characterize the nature of the changes, as we had difficulty capturing surges in these animals when sampled on presumptive proestrus. Therefore, we could not measure the dynamics (the precise time of onset, maximum amplitude, or duration) of their LH surges relative to those of WT mice. Nevertheless, with our modified sampling protocol, we did observe LH surges in GnRHR-Ctail females, which were reduced in amplitude. It is unlikely that this contributed to their subfertility, however, as there are several mouse models with reduced LH surge amplitudes that do not exhibit fertility defects (e.g., *Toufaily et al., 2020*; *Herbison et al., 2008*). Moreover, the amplitude of the surge varies dramatically between mice within a given strain (*Czieselsky et al., 2016* and our unpublished observations). Though we only detected LH surges in ~50% of GnRHR-Ctail mice, it is unlikely that they were truly blocked or absent in half of the animals. In the fertility trial, only one of eight animals was sterile. Similarly, in only one of six GnRHR-Ctail mice did we fail to observe corpora lutea in their ovaries. Thus, the complete absence of LH surges appears to be a rare event in these mice, most likely explained by inadequate FSH-stimulated follicle development and estrogen positive feedback. The cause of the variable (and low) penetrance of the infertility phenotype is presently unclear, but the animals were notably on a mixed genetic background.

The blunted LH surges in GnRHR-Ctail mice may derive, at least in part, from homologous receptor desensitization. The effects of adding Ctails to mammalian GnRHRs have been thoroughly investigated

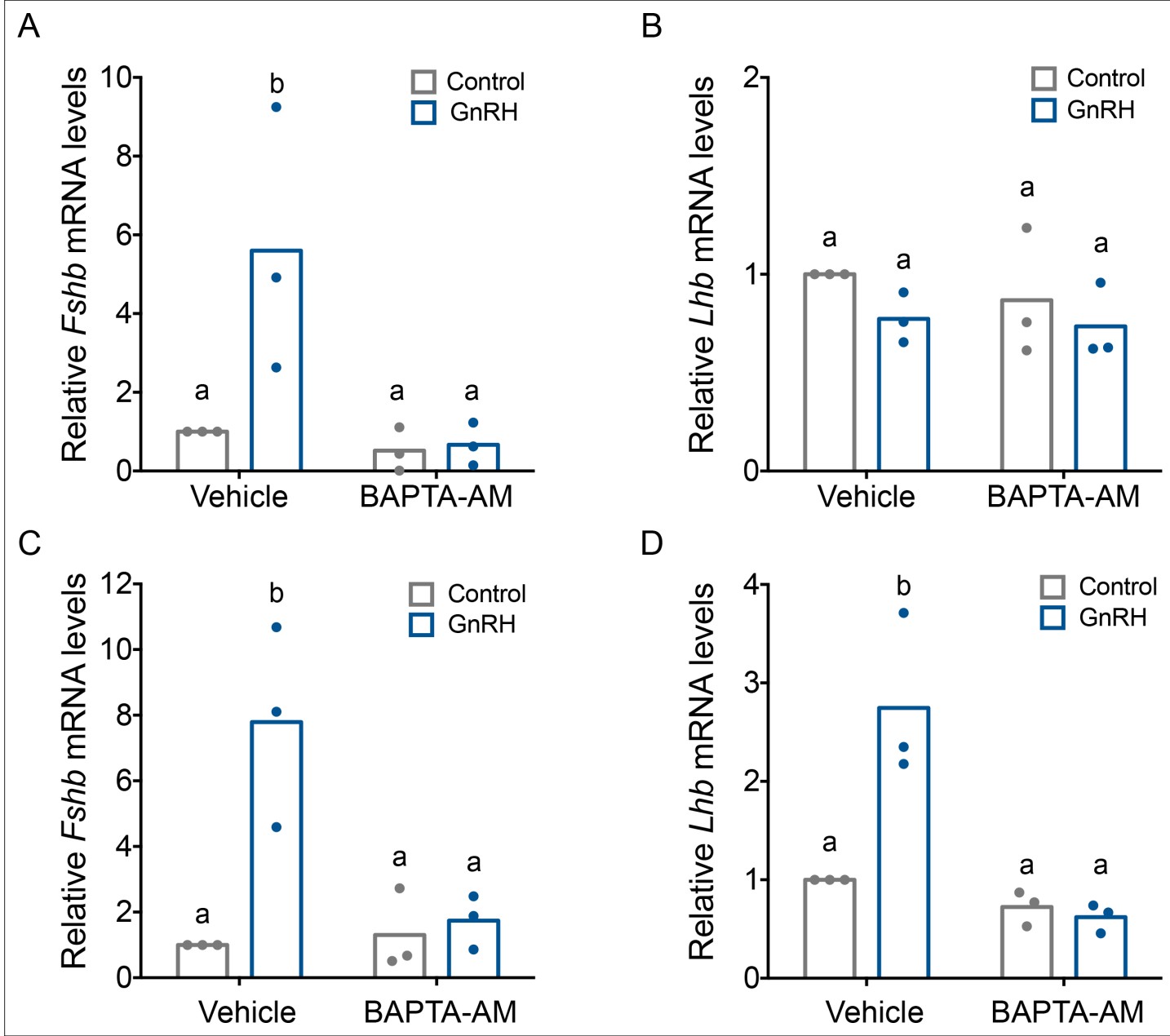

**Figure 9.** Gonadotropin-releasing hormone (GnRH)-induced *Fshb* expression is calcium dependent in homologous LβT2 cells. Relative *Fshb* (**A–C**) and *Lhb* (**B–D**) expression in LβT2 cells treated with vehicle (dimethyl sulfoxide, DMSO) or 20 µM BAPTA-AM for 20 min followed by treatment with water (control) or low (**A, B**) or high GnRH (10 nM) pulse frequency (**C, D**). Gene expression was assessed by RT-qPCR and normalized to the reference gene ribosomal protein L19 (*Rpl19*). Data shown are from three independent experiments. The bar heights reflect group means. Data were analyzed with two-way analyses of variance (ANOVAs), followed by post hoc Tukey test for multiple comparisons. Bars with different letters differed significantly. Panel A: control (vehicle) vs. GnRH (vehicle) p = 0.0193; control (vehicle) vs. control (BAPTA-AM) p = 0.9981; control (vehicle) vs. GnRH (BAPTA-AM) p = 0.9933; GnRH (vehicle) vs. GnRH (BAPTA-AM) p = 0.0122. Panel B: control (vehicle) vs. GnRH (vehicle) p = 0.5389; control (vehicle) vs. control (BAPTA-AM) p = 0.8495; control (vehicle) vs. GnRH (BAPTA-AM) p = 0.4184; GnRH (vehicle) vs. GnRH (BAPTA-AM) p = 0.9951. Panel C: control (vehicle) vs. GnRH (vehicle) p = 0.0072; control (vehicle) vs. control (BAPTA-AM) p > 0.9999; control (vehicle) vs. GnRH (BAPTA-AM) p = 0.9964; GnRH (vehicle) vs. GnRH (BAPTA-AM) p = 0.0143. Panel D: control (vehicle) vs. GnRH (vehicle) p = 0.0052; control (vehicle) vs. control (BAPTA-AM) p = 0.8627; control (vehicle) vs. GnRH (BAPTA-AM) p = 0.7190; GnRH (vehicle) vs. GnRH (BAPTA-AM) p = 0.0015.

The online version of this article includes the following figure supplement(s) for figure 9:

**Figure supplement 1.** Gonadotropin-releasing hormone (GnRH)-induced *Fshb* and *Lhb* expression does not depend on calcium entry via L-type channels in homologous LβT2 cells.

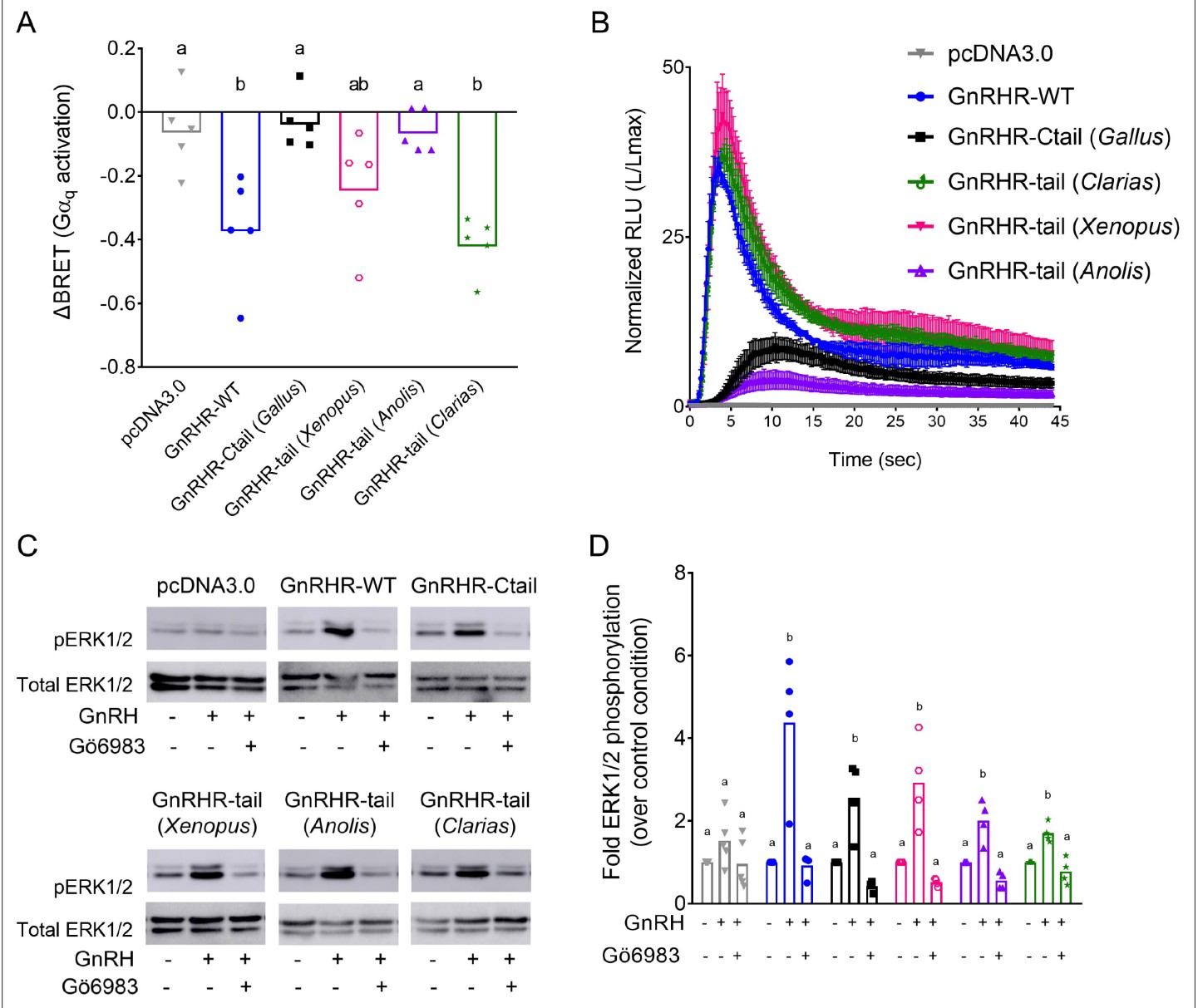

**Figure 10.** Disruption of gonadotropin-releasing hormone (GnRH) signaling via chimeric GnRHRs depends on the sequence of the Ctail. (**A**) HEK 293 cells were transfected with the G$_q$ biosensor and expression vectors for the indicated chimeric GnRHRs. Cells were treated with GnRH as in *Figure 6A*. BRET signals were read three times before and after GnRH stimulation. ΔBRET values are as described in *Figure 6*. Data are shown from five independent experiments. Bars heights reflect the group means. Data were analyzed with one-way analyses of variance (ANOVAs), followed by post hoc Tukey test for multiple comparisons. Different letters differed significantly. pcDNA3.0 vs. GnRHR-WT p = 0.0086; pcDNA3.0 vs. GnRHR-Ctail p = 0.9995; pcDNA3.0 vs. GnRHR-tail (*Xenopus*) p = 0.2477; pcDNA3.0 vs. GnRHR-tail (*Anolis*) p > 0.9999; pcDNA3.0 vs. GnRHR-tail (*Clarias*) p = 0.0021. (**B**) HEK 293 cells were transfected with the luminescence Obelin biosensor and expression vectors for the indicated chimeric GnRHRs. Cells were treated with GnRH and analyzed as in *Figure 7E*. The mean ± standard error of the mean (SEM) of four independent experiments is shown. (**C**) HEK 293 cells were transfected with the expression vectors for the indicated chimeric GnRHRs and treated and analyzed as in *Figure 7C*. The western blot shown is representative of four independent experiments. (**D**) The data exemplified in panel C were quantified and analyzed as in *Figure 7B*. Bar heights reflect group means. Data were analyzed with one-way ANOVA, for each receptor, followed by post hoc Tukey test for multiple comparisons. Bars with different letters differed significantly. pcDNA3.0: control (vehicle) vs. GnRH (vehicle) p = 0.4955; GnRH (vehicle) vs. GnRH (Gö6983) p = 0.1373. GnRHR-WT: control (vehicle) vs. GnRH (vehicle) p = 0.0027; GnRH (vehicle) vs. GnRH (Gö6983) p = 0.0023. GnRHR-Ctail: control (vehicle) vs. GnRH (vehicle) p = 0.0050; GnRH (vehicle) vs. GnRH (Gö6983) p = 0.0006. GnRHR-tail (*Xenopus*): control (vehicle) vs. GnRH (vehicle) p = 0.0046; GnRH (vehicle) vs. GnRH (Gö6983) p = 0.0010. GnRHR-tail (*Anolis*): control (vehicle) vs. GnRH (vehicle) p = 0.0036; GnRH (vehicle) vs. GnRH (Gö6983) p = 0.0003. GnRHR-tail (*Clarias*): control (vehicle) vs. GnRH (vehicle) p = 0.0043; GnRH (vehicle) vs. GnRH (Gö6983) p = 0.0007.

The online version of this article includes the following source data and figure supplement(s) for figure 10:

*Figure 10 continued on next page*

*Figure 10 continued*

**Source data 1.** Source data for *Figure 10*.

**Figure supplement 1.** Alignment of the Ctails from the GnRHRs in chicken, frog, lizard, and catfish.

in vitro. In most cases, these manipulations are associated with agonist-induced receptor phosphorylation, arrestin recruitment, and receptor internalization (*Vrecl et al., 2000*; *Willars et al., 1999*; *Pawson et al., 2008*; *Heding et al., 2000*; *Lin et al., 1998*). We similarly observed that the mouse-chicken chimeric GnRHR used here acquired the ability to recruit arrestin in response to GnRH. It is therefore likely that GnRHR-Ctail is rapidly internalized in response to agonist, but we did not assess this directly. Though arrestin recruitment to the chimera's Ctail did not appear to explain the altered calcium signaling or retained ERK activation in HEK 293 cells, we acknowledge that multiple GPCR/arrestin conformations, including differences between the receptor core and/or Ctail, can have distinct functions (*Cahill et al., 2017*; *Kumari et al., 2017*; *Baidya et al., 2020*; *Dwivedi-Agnihotri et al., 2020*). Regardless, if we were able to measure the duration of LH surges in GnRHR-Ctail females, we predict that it would be shorter than in WT mice. However, as most of these mice ovulated, the amplitude and duration of these surges were clearly sufficient. We recently reported that kisspeptin-54 induces surge-like LH release in juvenile mice. Although the duration of the LH increase is shorter than natural surges, these mice still ovulate efficiently (*Owen et al., 2021*). Thus, both the amplitude and duration of natural LH surges are greater than actually needed to induce ovulation in mice.

## Effects of the chicken Ctail on GnRH signaling

The reductions in gonadotropin production in GnRHR-Ctail mice indicate that the addition of the chicken Ctail altered GnRH signaling. In heterologous HEK 293 cells, GnRH stimulation of calcium mobilization was greatly impaired downstream of GnRHR-Ctail. This, in turn, appeared to be explained by attenuated activation of $G\alpha_q$ and reduced agonist stimulated inositol phosphate production. As GnRH induction of *Fshb* mRNA expression in homologous LβT2 cells is calcium dependent (*Lim et al., 2007*), it is possible that FSH deficiency in GnRHR-Ctail mice may result from alterations in calcium signaling. In contrast, GnRH induction of ERK1/2 phosphorylation is intact downstream of GnRHR-Ctail in HEK 293 cells. In gonadotropes, GnRH promotes ERK1/2 signaling via PKC, which in turn depends on diacylglycerol (DAG) more so than calcium (*Liu et al., 2005*). Though GnRH induction of DAG production was attenuated downstream of GnRHR-Ctail in HEK 293 cells, it was sufficient to activate PKC-ERK1/2 signaling. As GnRH regulation of *Lhb* expression is ERK1/2 dependent (*Bliss et al., 2009*), this may help explain how LH production was relatively unperturbed in gonad-intact GnRHR-Ctail mice.

The $G\alpha_q$ activation and, in particular, calcium mobilization impairments, downstream of GnRHR-Ctail in heterologous cells do not fully recapitulate changes in GnRH signaling in gonadotropes in GnRHR-Ctail mice. However, in both HEK 293 cells and gonadotropes, the GnRHR-Ctail induced a more sustained calcium profile. Gonadotropes possess L-type calcium channels, which are absent in HEK 293 cells (*Stutzin et al., 1989*; *Perez-Reyes et al., 1994*), though the latter do have endogenous calcium currents (*Berjukow et al., 1996*). GnRH-induced calcium oscillations in gonadotropes reflect both mobilization from ER stores and influx via voltage-dependent L-type channels. The calcium signaling (and defects therein) that we examined in HEK 293 cells is limited to mobilization from internal stores. Nevertheless, it is evident that GnRH-induced calcium oscillations also differ between gonadotropes of WT and GnRHR-Ctail mice. In WT pituitaries, we observe the previously reported heterogeneity of responses: oscillatory, biphasic, and transient (*Catt and Stojilković, 1989*). In contrast, GnRH stimulates a more homogenous calcium response in gonadotropes of GnRHR-Ctail mice and one that is not observed in WT animals. Relative to WT, gonadotropes of GnRHR-Ctail mice show sustained calcium oscillations, which extend well after the GnRH pulse. The mechanisms underlying this sustained activity are not clear but depend to some extent on influx of calcium via L-type channels. Regardless, the changes in calcium signaling from primarily transient intracellular release of calcium to a sustained influx of extracellular calcium may contribute to the observed reductions in FSH synthesis in GnRHR-Ctail mice.

In contrast, pulsatile LH secretion, which depends upon GnRH-induced calcium mobilization (*Tomić et al., 1994*), appears to be intact in GnRHR-Ctail males (note that we did not measure pulsatile LH secretion in females because of the high variability between estrous cycle stages [*Czieselsky et al.,*

*2016*] and the estrous cycle irregularities in GnRHR-Ctail mice). This 'normal' LH secretion may be more apparent than real, however. Exogenous GnRH stimulates less LH secretion in male GnRHR-Ctail than WT mice, despite their equivalent pituitary LH contents. GnRH is similarly less effective in stimulating LH release in GnRHR-Ctail females, but they also show marked decreases in pituitary LH content relative to WT, precluding a definitive interpretation of the results. LH secretion is blunted in both sexes following gonadectomy and at the time of the LH surge in females. Therefore, the alterations in GnRH stimulated calcium signaling may also affect LH secretion, which is most evident when GnRH pulse frequency or amplitude is enhanced.

It is possible that the phenotypes of GnRHR-Ctail mice are explained by reduced receptor expression rather than (or in additional to) altered receptor function. Indeed, *Gnrhr* mRNA levels are reduced in gonad-intact GnRHR-Ctail relative WT mice. We do not know if this translates into differences in GnRHR protein expression. Unfortunately, we were unable to identify reliable antibodies for measurement of GnRHR protein in the pituitary. We also could not validate GnRHR ELISAs used by others (*Odle et al., 2018*) (data not shown). In vitro ligand binding assays in pituitaries from the two genotypes do not provide a viable alternative means for receptor protein quantification, as *Gnrhr* mRNA levels decrease dramatically in cultured cells relative to in vivo and the genotype difference in *Gnrhr* expression does not persist in culture (data not shown). Regardless, we hypothesize that the reduced *Gnrhr* mRNA levels in GnRHR-Ctail mice are themselves a consequence rather than a cause of altered GnRH signaling. Not only does GnRH positively regulate the expression of its own receptor (*Naik et al., 1985b*; *Naik et al., 1985a*), but the WT and Ctail forms of the murine GnRHR are expressed at equivalent levels when transfected in heterologous cells. Thus, there does not appear to be any inherent difference in the stability of WT and Ctail forms of the receptor.

## Evolutionary significance of the loss of the Ctail

Finally, in light of all of the results, it is tempting to speculate that the loss of the Ctail from the mammalian GnRHR may have conferred a selective advantage by augmenting G-protein coupling, leading to enhanced calcium mobilization, FSH production, folliculogenesis, and fertility. However, more recent phylogenetic analyses suggest that the loss of the Ctail may be an ancient event in jawed vertebrates, predating mammalian evolution (*Williams et al., 2014*). It is unclear what advantage this may have conferred when it first emerged and why it has only been retained in mammals and a small number of other vertebrates. We are limited in what we can conclude or interpret from the one mouse model we examined here. While adding the chicken Ctail decreased FSH production, this may not have been the case if we had instead fused the *Xenopus* or *Clarias* Ctails, which do not appear to perturb GnRH signaling in vitro. Therefore, the presence of a Ctail, in and of itself, does not necessarily impair or alter G-protein coupling to the GnRHR. The specific sequence of the tail is relevant. It could be informative to reconstruct ancestral GnRHRs (*Hochberg and Thornton, 2017*) and then examine the effects of removing their Ctails on signaling. Though challenging, this may ultimately provide more, or at least parallel, insight into the potential adaptive significance of the loss of the Ctail. Regardless, the data presented here demonstrate that LH surges are possible in mammals in the presence of a GnRHR with a disruptive Ctail and suggest that FSH synthesis is dependent upon the nature of GnRH-dependent calcium signaling in gonadotropes.

## Materials and methods
### Reagents

GnRH (LH releasing hormone, L7134), nimodipine (66085-59-4), paraformaldehyde (PFA, 158127), bovine serum albumin fraction V (BSA, 10735078001), Triton X-100 (9002-93-1), dimethyl sulfoxide (DMSO, 472301), anti-Flag antibody (F7425; RRID:AB_439687), EZview Red ANTI-FLAG M2 Affinity Gel (F2426; RRID:AB_2616449), and 3X FLAG Peptide (F4799) were obtained from Sigma-Aldrich (St-Louis, MO, USA). The PKC inhibitor Gö6983 (ab144414) was from Abcam (Cambridge, UK). BAPTA-AM (126150-97-8) was from Tocris (Bristol, UK). TRIzol reagent (15596026) was from Life Technologies (Carlsbad, CA, USA). ProLong Gold antifade (P36931), Pluronic F-127 20% solution (P3000MP), and Fluo4-AM (F1420) were from Thermo Fisher Scientific (Waltham, MA, USA). Deoxynucleotide triphosphates (dNTPs, 800-401-TL), fetal bovine serum (FBS, 080150), and Dulbecco's modified Eagle medium (DMEM, 319-005 CL) were from Wisent Inc (St-Bruno, QC, Canada). Oligonucleotides were

synthesized by Integrated DNA Technologies (Coralville, IA, USA). Polyethylenimine (PEI, 23966) was from Polysciences Inc (Warrington, PA, USA). Coelenterazine cp (10112) and Coelenterazine 400a (10125) were from Biotium (Fremont, CA, USA). Phospho-p44/42 MAPK (Thr202/Tyr204 pERK1/2; 9101S; RRID:AB_331646) and p44/42 MAPK (ERK1/2; 9102S; RRID:AB_330744) antibodies were from Cell Signaling Technologies (Danvers, MA, USA). Horseradish peroxidase (HRP)-conjugated goat anti-rabbit (170-6515; RRID:AB_11125142) and goat anti-mouse (170-6516; RRID:AB_11125547) anti-bodies were obtained from BioRad Laboratories (Hercules, CA, USA).

## Cell lines and transfections

LβT2 cells (*Alarid et al., 1996*) were provided by Dr. Pamela Mellon (University of California, San Diego, CA, USA). HEK 293 WT and *Arrb1;Arrb2* CRISPR knockout (KO) cells were provided by Dr. Inoue Asuka (Tokyo University, Sendai, Japan; *Alvarez-Curto et al., 2016*). All cells were maintained and grown in DMEM (4.5 g/l glucose, with L-glutamine and sodium pyruvate) containing 10% (vol/vol) FBS at 37°C in a 5% $CO_2$ atmosphere. Transfections of HEK 293 cells lines were performed using PEI transfection reagent in a mass ratio of 3:1 PEI to DNA.

## Plasmids

pGEM-T Easy was purchased from Promega (Wisconsin, USA; Cat # A1360). To generate the flag-tagged murine GnRHR-WT and GnRHR-Ctail expression vectors, the murine *Gnrhr* coding sequence was amplified by PCR from LβT2 cell cDNA using a forward primer introducing an *Eco*RI restriction site and omitting the endogenous translation initiation codon and a reverse primer introducing an *Xba*I restriction site (*Table 1*). The resulting fragment was digested with the indicated enzymes and ligated in-frame downstream of the flag tag coding sequence preceded by a translation initiation codon in pcDNA3.0, yielding Flag-GnRHR. To generate the Flag-GnRHR-Ctail vector, the stop codon in Flag-GnRHR was replaced with a *Cla*I restriction site by site-directed mutagenesis (QuikChange protocol). The Ctail coding sequence from the chicken *Gnrhr* gene (*Gallus gallus*; NP_989984) was amplified by PCR from chicken embryonic genomic DNA (provided by Dr. Aimee Ryan, McGill University) using primers incorporating *Cla*I sites at both ends (*Table 1*). This fragment was inserted into the *Cla*I site created at the end of the *Gnrhr* coding sequence.

To generate other chimeric receptors, the Ctail coding sequence from the frog *Gnrhr* gene (*Xenopus leavis*; accession number NM_001085707) was PCR amplified from HA-XGnRHR (*Finch et al., 2008*; provided by Dr. Craig McArdle, University of Bristol, Bristol, UK) using primers incorporating *Cla*I sites at both ends (*Table 1*), replacing the amino acids underlined in *Figure 10—figure supplement 1*. Ctail coding sequences from the *Gnrhr* of the lizard *Anolis carolinensis* (XP_003226613.1) and from the catfish *Clarias gariepinus* (adapted from the coding sequence of *Tachysaurus fulvidraco Gnrhr* [XM_027175679.1] based on the peptide sequence described in *Lin et al., 1998*) were synthesized as double stranded DNA by Twist Biosciences (San Francisco, CA), harboring *Cla*I sites downstream of adaptors added at both ends. These Ctails were PCR amplified using primers complementary to the adaptor sequences (*Table 1*), digested with *Cla*I, purified, and ligated into *Cla*I-digested dephosphorylated Flag-GnRHR-Ctail, from which the chicken Ctail was excised. All clones were confirmed by Sanger sequencing at GenomeQuébec, Montreal, Canada.

The polycistronic Gα$_q$ (*Namkung et al., 2016a*) and DAG (*Namkung et al., 2018*) biosensors, and β-arrestin-1-YFP (*Khoury et al., 2014*) constructs were provided by Dr. Stéphane Laporte (McGill University, Montréal, Canada). The luminescence obelin biosensor (*Quoyer et al., 2013*) was provided by Dr. Michel Bouvier (Université de Montréal, Canada).

## Targeting vector

To generate the downstream chromosomal arm (DCA), a 6.7 kb DNA fragment starting 1 kb upstream of murine *Gnrhr* exon 3 was amplified by PCR using the Expand Long Template PCR System (Roche, Basel, Switzerland) from 129SvEv genomic DNA using primers incorporating 5′ *Xma*I and 3′ *Not*I restriction enzyme sites (*Table 1*). The fragment was cloned into pGEM-T Easy. The stop codon in exon 3 was replaced with a *Cla*I restriction enzyme site by site-directed mutagenesis. The *Cla*I-flanked Ctail coding sequence from the chicken *Gnrhr* (also used for the Flag-GnRHR-Ctail construct described above) was inserted, and the correct orientation was verified by sequencing. The whole DCA containing the chimeric exon 3 was ligated between the *Xma*I and *Not*I sites in pKOII (*Bardeesy*

**Table 1.** Primers.

All primers are listed in 5' to 3' orientation.

| Expression vectors | |
| --- | --- |
| Gnrhr ORF (for) | CGGAATCGCTCACAATGCATCTCTTGAG |
| Gnrhr ORF (rev) | ACTCTAGATCTCCAAAGAGAAATACCCATATA |
| pcDNA3.0-GnRHR Stop to ClaI (for) | GACCCACTCATATATGGGTATTTCTCTTTGATCGATTAGAGGGCCCTATTCTA TAGTGTCACCTA |
| pcDNA3.0-GnRHR Stop to ClaI (rev) | TAGGTGACACTATAGAATAGGGCCCTCTAATCGATCAAAGAGAAATACCCATAT ATGAGTGGGTC |
| Chicken Ctail (for) | CGGATCGATCGTTTCGGGAGGACGTGCAA |
| Chicken Ctail (rev) | CGGATCGATTCAGCACACCGTGTTAACGG |
| Gnrhr STOP to ClaI (for) | TGCACCCACTCATATATGGGTATTTCTCTTGATCGATGGAGAACTACACAAGA ACTCAGATAGAAATAAG |
| Gnrhr STOP to ClaI (rev) | CTTATTTCTATCTGAGTTCTTGTGTAGTCTCCATCGATCAAAGAGAAATCACC ATATATGAGTGGGTCGA |
| Xenopus tail (for) | TTAAATCGATAAAGAGGACCTGCGATCATGGATCA |
| Xenopus tail (rev) | AATTATCGATTCAGAAGACTGATTGCATGGT |
| Adaptor primer (for) | GAAGTGCCATTCCGCCTGAC |
| Adaptor primer (rev) | ACTGAGCCTCCACCTAGCCT |

*Table 1 continued on next page*

*Table 1 continued*

| | |
|---|---|
| **Expression vectors** | |
| **Targeting vector** | |
| Gnrhr UCA (for) | CGGGGTACCTATAAACTCATTAGCTGATTCAAACTT |
| Gnrhr UCA (rev) | CGGCCCGGGCAGTTCTGACAGACTAGCCCCC |
| Gnrhr floxed region (for) | CGGCCCGGGCGATAACTTCGTATAATGTATGCTAAAGTTATCAGGATTCACCT CACCATGG |
| | |
| Gnrhr floxed region (rev) | CGGGTTTAAACCTACAAAGAGAAATACCCAT |
| Gnrhr DCA (for) | CGGCCCGGGCAGGATTCACCTCACCATGG |
| Gnrhr DCA (rev) | CGGGGCGGCCGCAATTGAAGATCACAGTGTTT |
| BGH PolyATail (for) | TAAGTTTAAACCGCTGATCAGC |
| BGH PolyATail (rev) | CGGCTCGAGCCATAGAGCCCACCGCATC |
| | |
| **Southern blot probes** | |
| 5′ Southern probe (for) | CTTCAACCCGCCCTCTAGT |
| 5′ Southern probe (rev) | AGCCGGTCTAAGAATCCTCTC |
| 3′ Southern probe (for) | CAAAGTGCCCACAGATTTTG |
| 3′ Southern probe (rev) | GCCTGGTGTTCTGAGAGACTG |
| | |
| **Genotyping** | |
| Gnrhr WT exon 3 (for) | CTCGGCTGAGAACGATAAAG |
| Gnrhr WT exon 3 (rev) | CCCATATATGAGTGGGTCGAA |
| Gnrhr Ctail exon 3 (Ctail for) | TTCGCTACCTCCTTTGTCGT |

*Table 1 continued on next page*

Table 1 continued

| Expression vectors | |
|---|---|
| Gnrhr Ctail Ctail (Ctail rev) | TGTTAACGGTTGTCCCATT |
| gDNA Gnrhr (for) | CATGGAGATCCTTGCTGACA |
| gDNA Gnrhr (rev) | CACCTGGGGGCTAGTCTGT |

| qPCR | |
|---|---|
| Cga (for) | TCCCTCAAAAAGTCCAQGAGC |
| Cga (rev) | GAAGAGAATGAAGAATATGCAG |
| Fshb (for) | GTGCGGGCTACTGCTACACT |
| Fshb (rev) | CAGGCAATCTTACGGTCTCG |
| Gnrhr (for) | CACGGGTTTAGGAAAGCAAA |
| Gnrhr (rev) | TTCGCTACCTCCTTTGTCGT |
| Lhb (for) | AGCAGCCGGCAGTACTCGGA |
| Lhb (rev) | ACTGTGCCGGCCTGTCAACG |
| Rpl19 (for) | CGGGAATCCAAGAAGATTGA |
| Rpl19 (rev) | TTCAGCTTGTGGATGTGCTC |

*et al., 2002*), 3′ of the *Frt*-flanked neomycin (neo) selection cassette. We used a two-step process to generate the upstream chromosomal arm (UCA) and the 'floxed' exon 3 regions. First, a genomic DNA fragment starting 1 kb upstream of exon 3 and terminating immediately after the stop codon in exon 3 was amplified by PCR using a 5′ primer introducing a *Xma*I restriction site and a *loxP* site, and a 3′ primer introducing a *Pme*I restriction site (*Table 1*). This amplicon, along with a *Pme*I–*Xho*I fragment comprising the bovine growth hormone polyA sequence (obtained by PCR from the pcDNA3.0 expression vector) were ligated in a three-part ligation between the *Xma*I and *Xho*I restriction sites of pBluescript II KS. To complete the UCA, a 3.6 kb fragment spanning exon 2 and terminating 1 kb upstream of exon 3 (the position of the upstream *loxP* site) was amplified by PCR using primers incorporating 5′ *Kpn*I and 3′ *Xma*I sites (*Table 1*) and joined to the *Xma*I–*Xho*I construct (in pBluescript II KS) described above. The whole UCA was then ligated into the *Kpn*I and *Xho*I restriction sites in the pKOII vector containing the DCA, 3′ of the diphtheria toxin A negative selection cassette. Sequencing was performed to ensure the integrity of the targeting vector and the absence of mutations in and around exons and splice junctions (GenomeQuébec, Montreal, Canada). The targeting vector was linearized with *Kpn*I, phenol–chloroform extracted, and resuspended at a final concentration of 1 μg/ μl in Tris–EDTA.

## Generation of mice

Twenty-five μg of linearized targeting vector were electroporated into 10 million R1 ES cells (129/SvEv derived) in triplicate, and each electroporated sample plated on primary mouse embryonic fibroblasts in two 10-cm dishes. The following day, culture media was supplemented with 200 μg/ml G418 for positive selection of clones incorporating the targeting vector. After 8 days of selection, 420 clones were picked manually under a dissecting microscope, dissociated in trypsin, and transferred to individual wells of 96-well plates. Cells were cultured for 5 days and then split into three separate plates. Two plates were frozen at −80°C after the addition of 10% DMSO. Cells in the remaining plate were grown to confluence. Genomic DNA was extracted, cleaned with a series of 75% ethanol washes and digested overnight with *Xma*I. Homologous recombination events were screened by Southern blot using sequential hybridization with 5′ and 3′ probes external to the homology arms (see *Table 1* for the primers used to generate the probes).

C57BL/6J blastocysts were microinjected with cells from two correctly targeted clones and transferred into pseudopregnant mothers at the Transgenic Core Facility of the Life Science Complex at McGill University. Resulting chimeric males were bred to C57BL/6J females and germline transmission of ES cell-derived DNA monitored by coat color. Brown pups were genotyped by PCR for the presence of the modified allele (denoted *Gnrhr*^CtailfloxNeo^) and later confirmed by Southern blot. The Neo cassette was removed in vivo by breeding *Gnrhr*^CtailfloxNeo/+^ mice to 'flp deleter' mice (*B6.129S4-Gt(ROSA)26Sor^tm1(FLP1)Dym^/RainJ*, obtained from The Jackson Laboratory; *Farley et al., 2000*). The resulting *Gnrhr*^Ctailflox/+^ mice were bred to *EIIa::Cre* transgenic mice (*B6.FVB-Tg(EIIa-cre)C5379Lmgd/J*, obtained from the Jackson Laboratory; *Lakso et al., 1996*) to yield *Gnrhr*^Ctail/+^ mice (genotyped using primers Exon3 and Exon3-Ctail in *Table 1*). *Gnrhr*^Ctail/+^ females and males were then crossed to generate WT (*Gnrhr*^+/+^) and Ctail (*Gnrhr*^Ctail/Ctail^) mice. Genotyping was verified by PCR using the gDNA *Gnrhr* primers (*Table 1*). All animal experiments in Canada were performed in accordance with institutional and federal guidelines and were approved by the McGill University Facility Animal Care Committee (DOW-A; protocol 5204). Mice were maintained on a 12:12 light/dark cycle (on/off at 7:00 am/7:00 pm) and fed ad libitium. Mouse studies conducted at the National University of Mexico were performed under an institutional protocol similar to the United States Public Health Service Guide for the Care and Use of Laboratory Animals, and according to the Official Mexican Guide from the Secretary of Agriculture (SAGARPA NOM-062-Z00-1999).

## Estrous cycle staging and fertility assessment

Estrous cyclicity was assessed in 6-week-old mice for 21 consecutive days as described in *Ongaro et al., 2020*. At 9 weeks of age, females were paired with WT C57BL/6 males (Charles River, Senneville, QC, Canada) for a 6-month period. Breeding cages were monitored daily and the frequency of delivery and number of pups per litter were recorded. Pups were removed from cages 14 days after birth.

## Reproductive organ analyses, gonadal histology, and sperm counts

Testes, seminal vesicles, ovaries, and uteri were collected from 10- to 12-week-old males and females (diestrus afternoon). Body and organ masses were measured on a precision balance. Ovaries were fixed in 10% formalin, paraffin-embedded, and sectioned continuously at 5-µm thickness per section. Sections mounted on slides and then stained with hematoxylin and eosin for antral follicle, preovulatory antral (Graafian) follicle, and corpora lutea counting as described in *Li et al., 2018*. For sperm counts, testes were homogenized in 10% DMSO in 1× phosphate-buffered saline (PBS) using a Polytron PT10-35. Heads of mature spermatozoa were counted using a Leica DM-1000 microscope (Leica Microsystems, Wetzlar, Germany).

## Gonadectomy

Ovariectomy (OVX), castration (Cast), and sham operations were performed on 10-week-old mice following McGill University standard operating procedures #206 and #207 (https://www.mcgill.ca/research/research/compliance/animals/animal-research-practices/sop), respectively, as described in *Schang et al., 2020*.

## Blood collection

Blood was collected by cardiac puncture 2 weeks postoperatively (on diestrus afternoon for sham-operated females). Blood was allowed to clot for 30 min at room temperature and centrifuged at 1000 × *g* for 10 min to collect serum. Sera were stored at −20°C until hormone assays were performed. For LH pulsatility assessment in 10-week-old males, 2 µl of blood were collected from the tail tip every 10 min over 6 hr, starting 3 hr before lights off. For the LH surge onset and profile in 10-week-old females, 2 µl of blood were collected from the tail tip every 20 min over 8 hr on proestrus (as assessed by vaginal cytology). For LH surge amplitude assessment, four blood samples (4 µl each) were collected from the tail tip over 11 consecutive days: at 10:00 am, and at 6:00, 7:00, and 8:00 pm. The surge amplitude was defined as the maximal concentration of LH measured on days determined to be proestrus by vaginal smears. For the GnRH-induced LH release experiment, 4 µl of blood were collected from the tail tip of 10- to 11-week-old females (diestrus afternoon) and males just prior to and 15-, 30-, and 60-min post-i.p. injection of 1.25 ng of GnRH per g of body mass, diluted in 0.9% NaCl. Prior to all tail tip blood collection, animals were acclimatized by massaging the tail daily for 2 weeks. Tail tip blood samples for LH analysis were immediately diluted (1:30) in 1× PBS containing 0.05% Tween (PBS-T), gently vortexed, and placed on dry ice. Diluted blood was stored at −80°C until assayed.

## Hormone analyses

Serum FSH and LH levels were determined in males at the Ligand Assay and Analysis Core at the University of Virginia Center for Research in Reproduction using the mouse/rat LH/FSH multiplex assay (detection limit: 2.4–300 ng/ml; intra-assay CV <10%). In females, serum FSH was measured by the MILLIPLEX kit (MPTMAG-49K, Millipore, MA, USA) following the manufacturer's instructions (minimal detection limit: 9.5 pg/ml; intra-assay CV <15%) and serum LH was measured using an in-house sandwich ELISA, as previously described (*Steyn et al., 2013*) (detection limit: 0.117–30 ng/ml; and intra-assay CV <10%). Whole blood LH levels from both males and females were also measured using the in-house sandwich LH ELISA.

## Gonadotropin pituitary content assessment

Pituitaries were collected from 12- to 13-week-old female (randomly cycling) and male mice, placed on dry ice, and manually homogenized in 300 µl cold 1× PBS. Homogenates were centrifuged at 13,000 rpm for 15 min at 4°C. Total protein concentration was measured using the Pierce BCA Protein Assay Kit (23225; Thermo Fisher Scientific) following the manufacturer's instructions.

For FSH content assessment, samples were diluted 1:50 and FSH levels were measured by the MILLIPLEX kit (females) or by RIA (males) at the Ligand Assay and Analysis Core at the University of Virginia Center for Research in Reproduction. For LH pituitary content, samples were diluted 1:1,000,000 in PBS-T, and LH levels were measured using the LH ELISA indicated above. FSH and LH values were normalized over total protein content per pituitary.

## GnRH treatment of LβT2 cells

LβT2 cells were plated at 650,000 cells/well in 12-well plates and cultured overnight. The next day, cells were starved for 16–18 hr in serum-free medium. Cells were then pretreated for 20 min with

BAPTA-AM (20 µM) and then stimulated with one pulse of 10 nM GnRH (hereafter referred to as low GnRH pulse frequency). Two hours post-GnRH stimulation, media was replaced with fresh media containing the BAPTA-AM and incubated for an additional 2 hr. For high GnRH pulse frequency treatment, cells were stimulated with 10 nM GnRH for 5 min, every 45 min for a total of 10 pulses in the presence of the BAPTA-AM (20 µM) or vehicle. The latter were also included between GnRH pulses.

## Reverse transcription and quantitative PCR

Pituitaries were collected 2-week postgonadectomy (on diestrus afternoon for sham-operated females), snap frozen in liquid nitrogen, and stored at −80°C. Total RNA from pituitaries and LβT2 cells was isolated with TRIzol following the manufacturer's instructions. Pituitaries were first homogenized in 500 µl TRIzol using a Polytron PT10-35 homogenizer. RNA concentration was measured by NanoDrop and 250 ng of RNA per sample were reverse transcribed as in *Bernard, 2004*. Two µl of cDNA were used as a template in 20 µl reactions for quantitative real-time PCR analysis on a Corbett Rotorgene 600 instrument (Corbett Life Science) using EvaGreen reagent master mix. Relative gene expression was determined using the $2^{-\Delta\Delta Ct}$ method (*Livak and Schmittgen, 2001*) with the housekeeping gene ribosomal protein L19 (*Rpl19*) as reference (primers in *Table 1*).

## BRET assays

HEK 293 cells were plated at a density of 400,000 cells/well in 6-well plates. The next day, cells were cotransfected with PEI with 1 µg GnRHR-WT or GnRHR-Ctail expression vector (or empty vector as control) along with 1 µg of a polycistronic $G\alpha_q$ biosensor (*Namkung et al., 2016b*) or DAG biosensor (*Namkung et al., 2018*). Twenty-four-hour post-transfection, cells were detached by manual pipetting, and plated on poly-D-lysine-coated 96-well white plates at a density of 50,000 cells/well. The next day, cells were washed twice with Tyrode's buffer (140 mM NaCl, 1 mM CaCl$_2$, 2.7 mM KCl, 0.49 mM MgCl$_2$, 0.37 mM NaH$_2$PO$_4$, 5.6 mM glucose, 12 mM NaHCO$_3$, and 25 mM HEPES, pH 7.5). Next, cells were loaded with 5 µM Coelenterazine 400 a for 5 min in the dark at room temperature, and signals were subsequently recorded by a Victor X light plate reader (Perkin Elmer Life Sciences) starting 10 s before and continuing 30 s after 100 nM GnRH (or vehicle) injection, at 0.33-ms intervals. Net BRET was calculated as the ratio of the acceptor signal (GFP10 515/30-nm filter) over the donor signal (RLucII, 410/80-nm filter). ΔBRET was calculated by subtracting the average of basal BRET signals from ligand-induced signals (*Sleno et al., 2017*). Experiments with the $G\alpha_q$ biosensor and *Anolis*, *Xenopus*, and *Clarias* chimeric receptors were conducted as above, with the exception that a Synergy 2 Multi-Mode Microplate Reader (Bio Tek) was employed. Acceptor and donor signals were read three times before and after 100 nM GnRH (or vehicle) injection, at 16-s intervals.

## Cell line protein extraction, immunoprecipitation, and western blotting

Cellular extracts from HEK 293 and LβT2 cell lines were isolated using RIPA lysis buffer (50 mM Tris–HCl, 150 mM NaCl, 10 mM EDTA, 1% Triton X-100) as described in *Turgeon et al., 2017*. Total protein concentration was measured using the Pierce BCA Protein Assay Kit (23225; Thermo Fisher Scientific) following the manufacturer's instructions. Fifteen to 30 µg of total protein extracts were resolved by 10% sodium dodecyl sulphate–polyacrylamide gel electrophoresis, and transferred onto nitrocellulose membranes (1060001, GE Healthcare). Membranes were blocked for 1 hr at room temperature with blocking solution (Tris-buffered saline with 0.05% Tween [TBS-T] containing 5% skim milk). To investigate ERK1/2 phosphorylation, membranes were probed with rabbit anti-phospho-ERK1/2 (1:1000) for 16–18 hr at 4°C. For receptor expression, extracts were first incubated with EZview Red ANTI-FLAG M2 Affinity Gel and eluted with 3X FLAG peptide following manufacturer's instructions. Membranes were incubated with rabbit anti-Flag (1:1000). Following three washes with TBS-T, membranes were further incubated with HRP-conjugated goat anti-rabbit antibody (1:5000) in blocking solution for 2 hr at room temperature. For assessment of total ERK1/2 expression, membranes were stripped with 0.3 M NaOH, washed and incubated with anti-ERK1/2 (1:1000) following the same procedure as above for phospho-ERK1/2. Blots were incubated in Western Lightning ECL Pro reagent (Perkin Elmer) and then exposed on HyBlot CL film (E3012, Denville Scientific) or with a digital GE Amersham Imager 600. Band intensities were measured in arbitrary units using Image J software (US National Institutes of Health, Bethesda, MD, USA). Phosphorylated-ERK1/2 values were normalized over total ERK1/2 values in the same lane.

## IP1 production

HEK 293 cells were seeded at 400,000 cells/well in 6-well plates. The next day, cells were transfected with 1 µg of GnRHR-WT or GnRHR-Ctail expression vectors using PEI transfection reagent in a mass ratio of 3:1 PEI to DNA. Twenty-four-hour post-transfection, cells were detached by manual pipetting and replated in 384-well low volume white plates (15,000 cells/well) and incubated for an additional 24 hr. Next, cells were washed and stimulated with 0, 10, or 100 nM GnRH for 30 min at 37°C. IP1 production was assessed using IP-ONE-Gq Kit (Cisbio, Codolet, France) following the manufacturer's instructions. Homogeneous Time-Resolved Fluorescence (HTRF) was measured using a Synergy 2 Multi-Mode Microplate Reader (BioTek) and the ratio was calculated following the manufacturer's instructions. Data are presented as relative HTRF, where values of stimulated conditions were normalized over the value of untreated GnRHR-WT expressing cells.

## Intracellular calcium mobilization in heterologous cells

HEK 293 cells were plated at a density of 400,000 cells/well in 6-well plates. The next day, cells were cotransfected with 1 µg of GnRHR-WT receptor or the indicated chimeric receptor expression vectors (or empty vector as control), and 1 µg Obelin biosensor, using PEI transfection reagent in a mass ratio of 3:1 PEI to DNA. Twenty-four-hour post-transfection, cells ($10^6$/ml) were washed with phenol-free, serum-free DMEM supplemented with 0.1% BSA (media), detached manually, and loaded with media containing 5 mM of Coelenterazine cp for 2 hr, shaking in the dark, at room temperature. Subsequently, 50,000 cells (in 50 µl) were plated per well in 96-well white microplates and 100 nM of GnRH (or vehicle) were injected using Synergy 2 Multi-Mode Microplate Reader (BioTek) or Victor X light plate reader (Perkin Elmer Life Sciences). Luminescence signals were recorded for 30 s, every 22 ms, and kinetic measurements were normalized over the maximal response (Lmax) obtained by lysing the cells with 0.1% Triton X-100.

## Calcium imaging in pituitaries

Whole pituitaries were dissected and incubated (37°C, 95% $O_2$ and 5% $CO_2$) for 30 min with the calcium sensor Fluo 4AM (InVitrogen; Eugene, OR, USA) at a final concentration of 22 µM in 0.1% DMSO (Sigma, St. Louis MO, USA), 0.5% pluronic acid F-127 (Sigma) in artificial cerebrospinal fluid (ACSF; 18 mM NaCl, 3 mM KCl, 2.5 mM $CaCl_2$, 25.2 mM $MgCl_2$, 2.5 mM $NaHCO_3$, 11 mM glucose, and 1.1 mM HEPES). The pituitaries were immobilized in a drop of 3% agar and placed on top of a Plexiglas chamber, which was then attached to the microscope stage and was continuously perfused (2 ml/min) with ACSF at room temperature. The pituitary was positioned to enable visualization of the ventral surface of the gland.

Baseline activity was recorded for 3.5 min while the sample was perfused with ACSF. To evaluate GnRH effects, 1-min baseline activity was recorded, followed by application of 10 nM GnRH (Luteinizing hormone-releasing hormone human acetate salt (LHRH), BACHEM H-4005.0025 1062179; Bubendorf, Switzerland) for 30 s followed by a washout period with ACSF for 1.5 min. GnRH and ACSF solutions were directly applied to the recording chamber by a gravity-fed perfusion system.

To evaluate the contribution of voltage-gated calcium channels, after 1 hr of recovery, tissue was incubated for 30 s with 20 µM nimodipine (ALOMONE LABS N-150 N150SM0250; Jerusalem, Israel) followed by a second application of 10 nM GnRH alone (*Figure 8—figure supplement 1*) or in combination with 20 µM nimodipine (*Figure 8—figure supplement 2*) for 30 s. Finally, to determine cell viability, high potassium solution (50 KCl mM, 120 NaCl mM, 10 HEPES mM, 2 $CaCl_2$ mM, pH 7.4) was applied for 30 s. For each condition, the numbers of animals and cells analyzed are indicated in the figure legends.

Image acquisition was performed with a cooled CCD camera (HyQ; Roper Scientific, Acton, MA, USA); 600 images sequences were acquired with each image taken with 200-ms exposure. The tissue was viewed with an epifluorescence microscope (Leica M205FA; Leica Microsystem; Wetzlar, Germany) equipped with a PlanAPO 2.0% (0.35 NA) objective lens. The excitation and suppression filters were BP 480/40 and BP 527/30, respectively.

Image sequences (1200 images; 200-ms exposure) were obtained from a given field of view, before, during, and after GnRH application and were saved in TIFF format. Movies were processed and analyzed with ImageJ macros (NIH) to obtain numerical values of fluorescence intensity corresponding to $[Ca^{2+}]_i$ changes. Every responsive cell was selected manually from the obtained recordings. Values

of fluorescence were corrected for photobleaching and normalized using Igor Pro (Wavemetrics Inc; Portland, OR, USA) with a semiautomatic routine written by Pierre Fontaneaud (Institute of functional Genomics, Montpellier, France) to obtain $\Delta F = F - F_0$ values, which were then plotted with a routine written by Leon Islas, Ph.D. (Medicine Faculty, UNAM, Mexico City) to visualize activity of each cell over time. GnRH responsive cells were selected based on whether fluorescence values changed over time following ligand application. All the cells analyzed responded to the depolarizing solution of high potassium.

## Immunofluorescence microscopy

HEK 293 cells were plated on poly-D-lysine (1 mg/ml)-coated coverslips, placed in a 24-well plate. The next day, cells were transfected with 300 ng GnRHR-WT or GnRHR-Ctail along with 200 ng of pYFP-Arrestin. Twenty-four- to 48-hr post-transfection, cells were treated with 100 nM GnRH or 1 µM of the GnRH analog Lucrin for 1, 5, or 20 min or left untreated as control. Cells were then fixed for 20 min with fresh 4% PFA and 0.2% Triton X-100. Coverslips were than mounted on glass slides using ProLong Gold. Fluorescence images were acquired with a Leica DM-1000 microscope with a ×40 objective, or confocal microscope (Leica SP5) with a 63 × 1.4 numerical aperture objective. Leica LAS AF image software was utilized for image acquisition.

## Cell surface expression with whole cell anti-Flag ELISA

HEK 293T cells were plated at a density of 40,000 cell/well on poly-D-lysine (1 mg/ml) coated 24-well plates. The next day, cells were transfected with 500 ng of empty vector, GnRHR-WT, or GnRHR-Ctail using PEI transfection reagent in a mass ratio of 3:1 PEI to DNA. Forty-eight-hour post-transfection, cells were fixed with fresh PFA (4%) for 20 min, washed three times gently with 1× PBS, and blocked for 2 hr with blocking solution (1× PBS containing 5% nonfat milk and 5% goat serum). Cells were incubated with rabbit anti-flag antibody (1:5000 in blocking solution) for 2 hr at room temperature. Next, cells were washed three times for 5 min each with 1× PBS, and further incubated for 1 hr with HRP-conjugated goat anti-rabbit antibody (1:5000 in blocking solution), followed by five washes for 5 min each with 1× PBS, and finally incubated in 500 µl of 3,3',5,5'-tetramethylbenzidine substrate for 15 min. The reaction was stopped by adding 2 N sulfuric acid. Absorbance was measured at 450 nm using a Biochrom Asys UVM 340 microplate reader.

## Statistical analyses

Statistical analyses were performed using GraphPad Prism software version 6 or 8 (San Diego, CA, USA, http://www.graphpad.com/), with the following exceptions. LH pulse data from males were deconvoluted using MatLab. For whole-gland intracellular calcium experiments, the AUC and MIF from the normalized values were extracted using R (version 3.5.1; https://www.R-project.org/) along with the following packages: dplyr, pracma, ggplot2, and ggpubr. The number of oscillations in gonadotropes identified with the oscillatory pattern of calcium mobilization was quantified automatically with the MathLab-based toolbox PeakCaller (*Artimovich et al., 2017*). The statistical significance of AUC, MIF, and number of oscillations (peaks) was tested in GraphPad using Wilcoxon rank-sum test when comparing means of two groups, or Wilcoxon signed-rank test when comparing responses of one cell to two stimuli. Statistical tests used, number of experiments, number of biological replicates, and p values are indicated in the figure legends. Results were considered statistically significant when $p < 0.05$.

## Acknowledgements

We thank Dr. Aimee Ryan and Dr. Stéphane Laporte from McGill University, Canada; Dr. Michel Bouvier from Institute for Research in Immunology and Cancer, Canada; Dr. Craig McArdle from Bristol University, UK; Dr. Inoue Asuka from Tokyo University, Japan; Dr. Pamela Mellon from University of California, USA for providing the indicated reagents. This work was supported by the Canadian Institutes of Health Research operating grants MOP-123447 and PJT-252739 to DJB; the Mexican National Counsel of Science and Technology operating grant CONACYT 273513 and the National University of Mexico operating grant PAPITT IN227416 to TF. TEH holds the Canadian Pacific Chair in Biotechnology.

## Additional information

### Funding

| Funder | Grant reference number | Author |
|---|---|---|
| Canadian Institutes of Health Research | MOP-123447 and PJT-252739 | Daniel J Bernard |
| Mexican National Counsel of Science and Technology | CONACYT 273513 | Tatiana Fiordelisio |
| National University of Mexico | PAPITT IN227416 | Tatiana Fiordelisio |
| McGill University | Canadian Pacific Chair in Biotechnology | Terence E Hébert |

The funders had no role in study design, data collection and interpretation, or the decision to submit the work for publication.

### Author contributions

Chirine Toufaily, Conceptualization, Formal analysis, Investigation, Methodology, Project administration, Writing – original draft, Writing – review and editing; Jérôme Fortin, Conceptualization, Formal analysis, Investigation, Methodology, Writing – review and editing; Carlos AI Alonso, Formal analysis, Investigation, Writing – original draft, Writing – review and editing; Evelyne Lapointe, Frederik Steyn, Terence E Hébert, Methodology, Writing – review and editing; Xiang Zhou, Dominic Devost, Aylin C Hanyaloglu, Investigation, Methodology, Writing – review and editing; Yorgui Santiago-Andres, Formal analysis, Investigation, Methodology, Writing – review and editing; Yeu-Farn Lin, Yiming Cui, Ying Wang, Investigation, Writing – review and editing; Ferdinand Roelfsema, Formal analysis, Investigation, Writing – review and editing; Tatiana Fiordelisio, Formal analysis, Investigation, Methodology, Writing – original draft, Writing – review and editing; Derek Boerboom, Methodology, Supervision, Writing – review and editing; Daniel J Bernard, Conceptualization, Formal analysis, Funding acquisition, Investigation, Methodology, Project administration, Supervision, Writing – original draft, Writing – review and editing

### Author ORCIDs

Yorgui Santiago-Andres ![ORCID] http://orcid.org/0000-0001-7343-7746
Aylin C Hanyaloglu ![ORCID] http://orcid.org/0000-0003-4206-737X
Tatiana Fiordelisio ![ORCID] http://orcid.org/0000-0002-9282-1476
Daniel J Bernard ![ORCID] http://orcid.org/0000-0001-5365-5586

### Ethics

All mouse experiments in Canada were performed in accordance with institutional and federal guidelines and were approved by the McGill University Facility Animal Care Committee (DOW-A; protocol 5204). Mouse studies conducted at the National University of Mexico were performed under an institutional protocol similar to the United States Public Health Service Guide for the Care and Use of Laboratory Animals, and according to the Official Mexican Guide from the Secretary of Agriculture (SAGARPA NOM-062-Z00-1999).

### Decision letter and Author response

Decision letter https://doi.org/10.7554/eLife.72937.sa1

## Additional files

### Supplementary files

• Transparent reporting form

### Data availability

All data generated or analysed during this study are included in the manuscript and supporting files.

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
