## [Editor Report]

The authors have studied the effects of addition of a C tail to mammalian GnRH receptor. This is very well conducted study with very nicely designed experiments and appropriate conclusions. This is an interesting study that was well conducted and written. It is an important observation on how evolution of the GnRH receptors have contributed to reproductive processes and hence an important study in the area of reproductive biology.

---

## [Decision Letter]

**Decision letter after peer review:**

Congratulations, we are pleased to inform you that your article, "Addition of a carboxy terminal tail to the normally tailless gonadotropin-releasing hormone receptor impairs fertility in female mice", has been accepted for publication in eLife.

*Reviewer #1:*

In this study, the authors have investigated the effect of introducing the c terminal to the GnRHR on the reproductive functions of mice. The C terminal tail that is present in chicken GnRH receptor into the mammalian GnRH receptor causes significant reduction in the overall gonadal function.

This is a very well conducted study. The experiments are very well designed and executed, and manuscript is well written.

The authors have studied the effect of addition of a Ctail to mammalian GNRHR which responsible for agonist induced receptor internalization and desensitization caused by phosphorylation of the Ctail, and recruitment of b arrestin.

The entire process of characterization of the Ctail GnRH receptor and its responsiveness has been demonstrated quite well and the effects of the Ctail receptor on the reproduction in both female and male mice have been very clearly established. It is evident that while both gonadotropins are affected by the Ctail receptor, it is the decrease in FSH that seem to be affecting the gonadal function, particularly in the females. There is a small decrease in FSHbeta message, as well as decrease in GnRHR expression. Is there a decrease in the total gonadotropin content in the pituitaries? It is also essential to establish the GnRHR density and affinity of the receptor. However, it is understandable that GnRH binding assays are not very reproducible. It is essential that some quantification of the receptor is necessary.

There is a decrease in antral follicles and corpora lutea in Ctail animals accounting for subfertility. However, the authors have not described in detail the effects on the ovarian histology. Was there any increase in the atretic follicles? With decrease in FSH levels that may be a real possibility. The authors should stain the ovarian section for apoptotic cells.

The more interesting is the effects on testicular function that have not been documented adequately in the manuscript. There is certainly decrease in serum FSH compared to the LH levels. Although the authors have shown that there is decrease in sperm numbers, there should investigate effects on the germ cell populations by carrying out flow cytometric analysis of the testicular germ cells. Further, they should stain the testicular sections for the apoptotic cells. However, these experiments can be carried out in the future studies.

*Reviewer #3:*

This is an interesting study to understand the function of the C tail, normally found in the GNRH receptor of the non-mammalian vertebrates. To address this the authors have employed a variety of approaches which include generation of mice with a GNRH receptor to which bird's C tail has been fused. Using this model, a variety of parameters have been compared between wild type and animals with GNRH receptor with Bird's c tail is fused. These include fertility status, serum gonadotrpin levels, subunit mRNA levels, and more importantly LH surge levels, (as on the main function of GnRH is to induce LH surge) as well as GNRH stimulated LH levels, and GnRH stimulated calcium mobilization and GNRH stimulated DAG in heterologous cells. It was found that none of the above parameters were affected significantly. These studies permit the authors to conclude that the C terminal tail does not provide any additional advantage to the mammalian system. In addition, the authors conclude that the loss of the C tail may have been an ancient event in jawed vertebrates although it does not offer any special advantage in mammals. This I feel is something similar to the extended C terminal of hCG in humans though both normal LH and CG with additional C terminal bind to same receptor. This extension in CG beta has been attributed to frame shift in the codon. The studies reported in the paper have been carried out very well and the conclusions drawn are well supported by the results obtained. In conclusion I recommend that the manuscript can be accepted in the present form.

I recommend the manuscript can be accepted in the present form.